# Towards the Mysteries of Convergent Interaction Representations through DNNs

## Abstract

This paper explores whether the generalization power of DNNs can be understood through the generalizability of interactions between input variables they encode, which is one of the central challenges in the field of symbolic generalization. Specifically, we propose and validate the following two propositions. First, we validate that the representation convergence of interactions and the overfitting degree of a DNN are strongly negatively correlated. Second, we demonstrate that proactively enhancing interaction convergence can effectively mitigate overfitting. Our results show that, apart from our interaction-level consistency, other forms of representational consistency do not effectively mitigate a DNN's overfitting. Furthermore, eliminating non-convergent interactions also successfully improves the proportion of interactions that generalize to testing samples.

## 1 Introduction

Previous studies have explored the generalization power of deep neural networks (DNNs) from different perspectives, including generalization bounds on high-dimensional feature spaces (Neyshabur et al., 2017; Bartlett & Mendelson, 2002), the flatness of the loss landscape (Hochreiter & Schmidhuber, 1997), and neural tangent kernel (NTK) theory (Jacot et al., 2018).

However, the understanding of AI models' generalization has progressively evolved toward more precise and nuanced perspectives. A fundamental question remains open in the field of model generalization: **can the generalization power of an "entire" DNN be decomposed into the generalizability or representation quality of the "compositional" inference patterns it encodes?**

**Background: symbolic generalization.** The emerging research direction of symbolic generalization (Ren et al., 2023; Li & Zhang, 2023a; Ren et al., 2024a;b) has undertaken many pioneering studies to answer the above question. Unlike mechanistic interpretability studies (surveyed in Appendix A), symbolic generalization does not explain specific neurons in a DNN. Instead, Ren et al. (2024a) have proven the counterintuitive phenomenon that **the highly complex inference patterns encoded by a DNN can be rigorously explained as a small set of symbolic interactions between input variables.** As shown in Figure 1, dozens of interactions can be extracted from an input sample. Each interaction represents an AND relationship (or an OR relationship) between input variables that is equivalently encoded by the DNN. For example, an AND interaction between four words $T = \{ice, melted, due, to\}$ is activated if and only if all four words co-occur in the input prompt. Once activated, this interaction contributes a numerical effect $I_T^{\text{AND}} = 1.34$ that boosts the model's confidence in generating the token "*temperature*." It has been proven (Ren et al., 2024a) that **such numerical effects based on AND-OR interactions can accurately predict network outputs on exponentially many samples**, thereby ensuring the rigor of the explanation. Please see the **video demo** in the supplementary material for symbolic generalization.

**Our work.** Therefore, in this study, we further investigate another fundamental question for symbolic generalization: *can the generalization power[1] of a DNN be attributed to the generalizability of symbolic interactions it encodes?* To this end, preliminary studies have revealed specific characteristics of interactions, *e.g.,* the low generalizability of complex interactions (Zhou et al., 2024). In particular, we follow (Chen et al., 2024) to **define the generalizability of an interaction as the**

---

[1]To simplify this research, we provisionally follow the common consensus (Zhang et al., 2019) to use the gap between training and testing losses to measure a model's overfitting (non-generalization) level.

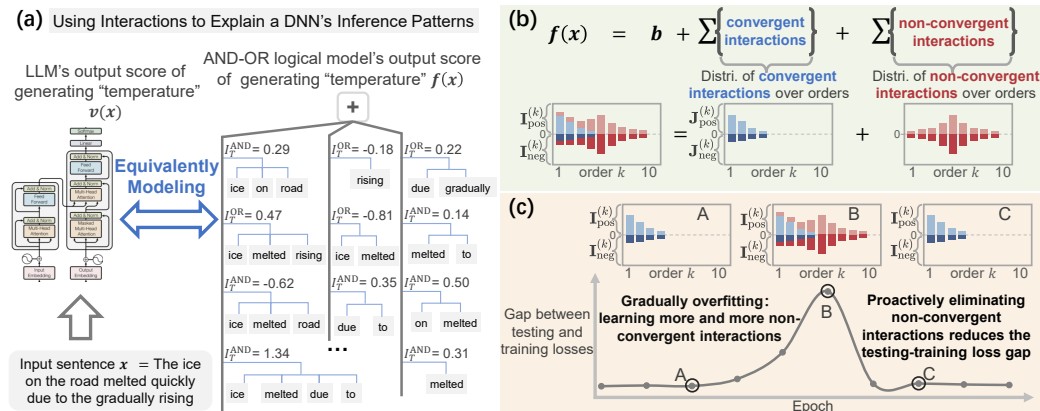

Figure 1: (a) It has been proven by (Ren et al., 2024a) that the complex inference patterns of a DNN can be faithfully represented by a logical model $f(x)$ composed of sparse AND-OR interactions. **Please see Figure 8 for examples of an AND-OR logical model that explains LLMs.** (b) Some interactions disentangled from the output are convergent over DNNs, while others are not convergent over DNNs. (c) We find that the convergence of interaction representations and a DNN's overfitting are closely intertwined, with the two phenomena mutually influencing each other.

**representation convergence of the interaction over different DNNs.** This is because if different DNNs trained for the same task, even with diverse architectures, are usually optimized toward a similar set of interactions, then these converged interactions are considered generalizable.

However, previous descriptive studies have not established a rigorous link between the overall generalization[1] of DNNs and the generalizability/convergence of interactions. Therefore, *we combine interventional and descriptive approaches to investigate their explicit relationship.* In this way, we aim to validate the following two interrelated propositions:

**Proposition 1, descriptive study:** The representation convergence of interactions and the overfitting degree of a DNN[1] are strongly negatively correlated. *I.e.,* when several DNNs are trained on the same task, the representation convergence of interactions across these networks keeps increasing before the overfitting, and gradually declines once these DNNs enter the overfitting phase[1].

**Proposition 2, interventional study:** Enhancing the representation convergence of interactions effectively mitigates the DNN's overfitting[1]. Specifically, proactively eliminating non-convergent interactions during training substantially reduces the training-testing loss gap.

The above two propositions have been consistently validated across DNNs with different architectures and a variety of tasks. In particular, we find that among various forms of representational consistency across DNNs, only enhancing the representation convergence of interactions can effectively mitigate a DNN's overfitting[1]. Together, these two propositions provide complementary evidence that a DNN's overfitting is strongly related to the convergence of its symbolic interactions. Validating them from both descriptive and interventional perspectives strengthens the foundation of the symbolic generalization theory.

**Theoretical value.** First, this paper builds upon and substantially extends findings in prior empirical studies. For example, Zhou et al. (2024) have found that high-order interactions generalize less than low-order ones, and Ren et al. (2024b) have found that mutually offsetting interactions also exhibit weak generalizability. Our research extends these conclusions by demonstrating that eliminating non-convergent interactions primarily removes high-order and mutually offsetting interactions, thereby explaining the root cause of mitigating a DNN's overfitting.

Second, we find that eliminating non-convergent interactions not only reduces the testing-training cross-entropy loss gap of DNNs, but also increases the ratio of interactions that generalize to testing samples. To the best of our knowledge, this represents the first effort to enhance the generalizability of detailed inference patterns in a DNN.

Third, our findings provide significant insights into identifying the precise point of early stopping in DNNs. Moreover, eliminating non-convergent interactions can also slightly improve the performance (testing accuracy) of models in practice. Please refer to Appendix E for detailed results.

## 2 TWO PROPOSITIONS FOR SYMBOLIC GENERALIZATION

### 2.1 PRELIMINARIES: INTERACTIONS

To formalize our ideas, we first define how a DNN's inference patterns can be expressed as symbolic interactions. Let us consider a DNN $v$ and an input sample $\boldsymbol{x} = [x_1, x_2, \ldots, x_n]^T$, which contains $n$ input variables[2] and is indexed by $N = \{1, 2, \ldots, n\}$. Let $v(\boldsymbol{x}) \in \mathbb{R}$ denote a scalar output of the DNN, and people may define different outputs $v(\boldsymbol{x})$ for different tasks, *e.g.,* the widely used classification confidence score in multi-class classification (Deng et al., 2022), as follows.

$$v(\boldsymbol{x}) \stackrel{\text{def}}{=} \log \frac{p(y = y^*|\boldsymbol{x})}{1 - p(y = y^*|\boldsymbol{x})} \in \mathbb{R}, \tag{1}$$

where $p(y = y^*|\boldsymbol{x})$ denotes the predicted probability of classifying $\boldsymbol{x}$ to the ground-truth label $y^*$.

**Problem setting:** Explaining the detailed inference patterns of a DNN is a core challenge in the field of *symbolic generalization* (Ren et al., 2023; Li & Zhang, 2023a; Ren et al., 2024a;b). In this theory, a logical model $f$ is introduced to explain the detailed inference patterns in the DNN $v$, and it uses two seemingly conflicting requirements to ensure the faithfulness of the explanation. **(1) Fidelity requirement:** The logical model $f$ must accurately predict the DNN's outputs across a vast number of input samples in a sufficiently large set $\Psi$. **(2) Conciseness requirement:** The logical model $f$ is expected to encode sufficiently simple logic, thereby facilitating concise explanations, as follows.

$$\forall \boldsymbol{x}' \in \Psi, \quad f(\boldsymbol{x}') = v(\boldsymbol{x}'), \quad \text{subject to} \quad \text{complexity}(f) \leq M, \tag{2}$$

where $M$ is an upper bound on the complexity of the logical model $f$.

Specifically, **the model $f$ is implemented as the following AND-OR logical model, which encodes a set of AND-OR interaction logic as illustrated in Figure 1.** *Please refer to the **video demo** in the supplementary material, which illustrates the core idea of the explanation theory. Please see Figure 8 for examples of an AND-OR logical model that explains LLMs.*

$$\forall \boldsymbol{x}' \in \Psi, \quad f(\boldsymbol{x}') \stackrel{\text{def}}{=} \sum_{T \in \Omega^{\text{AND}}} \underbrace{I_T^{\text{AND}} \cdot \mathbb{1}\left(\substack{\boldsymbol{x}' \text{ triggers AND} \\ \text{relation between } T}\right)}_{\text{an AND interaction}} + \sum_{T \in \Omega^{\text{OR}}} \underbrace{I_T^{\text{OR}} \cdot \mathbb{1}\left(\substack{\boldsymbol{x}' \text{ triggers OR} \\ \text{relation between } T}\right)}_{\text{an OR interaction}} + b. \tag{3}$$

• *The trigger function* $\mathbb{1}\left(\substack{\boldsymbol{x}' \text{ triggers AND} \\ \text{relation between } T}\right) \in \{0, 1\}$ *represents an AND relation between a subset* $T \subseteq N$ *of input variables.* It returns 1 if all variables in $T$ are present (not masked[3]) in $\boldsymbol{x}'$; otherwise, it returns 0. $I_T^{\text{AND}} \in \mathbb{R}$ is a scalar weight. $b$ is a scalar bias.

• *The trigger function* $\mathbb{1}\left(\substack{\boldsymbol{x}' \text{ triggers OR} \\ \text{relation between } T}\right) \in \{0, 1\}$ *represents an OR relation between a subset* $T \subseteq N$ *of input variables.* It returns 1 whenever any variable in $T$ present (not masked[3]) in $\boldsymbol{x}'$; otherwise, it returns 0. $I_T^{\text{OR}} \in \mathbb{R}$ is a scalar weight. $\Omega^{\text{AND}}$ and $\Omega^{\text{OR}}$ represent the set of AND interactions and the set of OR interactions extracted by the DNN $v$ from the input sample $\boldsymbol{x}'$.

**First, the fidelity requirement is satisfied by the following *universal matching property* in Theorem 1**. It shows that, with a specific setting of weights $I_T^{\text{AND}}$ and $I_T^{\text{OR}}$, the logical model's output $f(\cdot)$ can always accurately mimic the DNN's output $v(\cdot)$, no matter how the input sample $\boldsymbol{x}$ is augmented by enumerating all its $2^n$ masked states. The sample set $\Psi = \{\boldsymbol{x}_S | S \subseteq N\}$ is thus implemented as $2^n$ masked states of the input sample $\boldsymbol{x}$, which is sufficiently large. $\boldsymbol{x}_S$ denotes a masked input sample that retains only the input variables in $S$, while those in $N \setminus S$ are masked[3].

**Theorem 1 (Universal matching property**, proven in (Chen et al., 2024) and Appendix D). *Given a DNN $v$ and an input sample $\boldsymbol{x}$, let the scalar weights $I_T^{AND}$ and $I_T^{OR}$ in the logical model be set as* $\forall T \subseteq N$, $I_T^{AND} = \sum_{L \subseteq T}(-1)^{|T|-|L|}u_L^{AND}$, $I_T^{OR} = -\sum_{L \subseteq T}(-1)^{|T|-|L|}u_{N \setminus L}^{OR}$, *where* $u_L^{AND} = 0.5 \cdot v(\boldsymbol{x}_L) + \gamma_L$ *and* $u_L^{OR} = 0.5 \cdot v(\boldsymbol{x}_L) - \gamma_L$. *$\{\gamma_L\}$ is a set of learnable parameters. The scalar bias is set as $b = v(\boldsymbol{x}_\emptyset)$ and the sample set $\Psi$ is set as $\Psi = \{\boldsymbol{x}_S | S \subseteq N\}$. Then, no matter how we set $\{\gamma_L\}$, we always have*

$$\forall \boldsymbol{x}' \in \Psi, \quad f(\boldsymbol{x}') = v(\boldsymbol{x}'). \tag{4}$$

---

[2] For image classification tasks, input variables are often considered as patches in an image. In language generation tasks, each input variable typically embodies the embedding vector of a token.

[3] To mask an input variable in $S$, the variable is replaced with a baseline value, which is commonly defined as the average value of the variable across multiple input samples (Dabkowski & Gal, 2017).

**The extraction of AND-OR interactions** is introduced in (Li & Zhang, 2023b). It uses a LASSO-like objective function $\min_{\{\gamma_L\}} \sum_T \left( |I_T^{\text{AND}}| + |I_T^{\text{OR}}| \right)$ to learn sparse interactions by optimizing the learnable parameters $\{\gamma_L\}$. *The pseudocode for computing $I_T^{AND}$ and $I_T^{OR}$ is provided in Appendix K.*

**Second, the conciseness requirement is satisfied by *the sparsity property* of interactions.** Specifically, under three common conditions (see Appendix I), Ren et al. (2024a) have proven that, a well-trained DNN usually encodes only $\mathcal{O}(n^\kappa/\tau)$ salient interactions, where $\tau$ is a small positive scalar to identify salient interactions, and $\kappa \in [0.9, 1.2]$. Salient interactions are defined as those with significant effects *s.t.* $|I_T^{\text{AND}}| > \tau$ or $|I_T^{\text{OR}}| > \tau$. All other interactions have negligible effects.

Therefore, the above requirements ensure that we can construct a sufficiently concise logical model $f$ with only a few salient interactions by defining $\Omega_{\text{sparse}}^{\text{AND}} = \{T \subseteq N : |I_T^{\text{AND}}| > \tau\}$ and $\Omega_{\text{sparse}}^{\text{OR}} = \{T \subseteq N : |I_T^{\text{OR}}| > \tau\}$. **This sufficiently concise logical model $f$ can accurately approximate the DNN's outputs over the exponentially many masked states of the input sample $x$ in $\Psi$, and can therefore be regarded as an objective explanation of the DNN.**

**Order/complexity of interactions.** The order (or complexity) of an interaction is defined as the number of input variables in the set $T$, *i.e.,* $\text{order}(T) = |T|$.

## 2.2 PROPOSAL OF TWO PROPOSITIONS

The universal matching property and the sparsity property in Section 2.1 have demonstrated that the AND-OR interactions extracted from a DNN can faithfully represent the primitive inference patterns used by the DNN. This naturally leads to an important question: whether the generalization power of an entire DNN is determined by the representation quality of interactions in the DNN. Here, the representation quality is also termed the interactions' *generalizability*.

Along this research direction, many studies (Ren et al., 2023; 2024a;b; Zhou et al., 2024; Liu et al., 2023a) have emerged in recent years. However, previous studies have largely been limited to **descriptive analyses** of interaction properties, as demonstrated by the following observations:

• Zhou et al. (2024) found that high-order interactions generalized less than low-order ones.

• Liu et al. (2023a) found that high-order interactions were more sensitive to feature noise and harder to learn than low-order ones.

• Chen et al. (2024) proposed a method to extract interactions shared by different DNNs.

**Exploring potential mechanistic determinants of the link between interaction generalizability and a DNN's overfitting.** Therefore, in this study, we move beyond **descriptive analyses** toward **interventional approaches**, so that we can reveal more explicit mechanistic links between interaction generalizability and overfitting. Formal definitions are given as follows.

**(1)** *Interaction generalizability* **is defined in terms of representation convergence**, following (Chen et al., 2024). Specifically, the representation convergence means that when several DNNs are trained for the same task, different DNNs usually encode similar sets of salient interactions. *The interactions shared by different DNNs are considered generalizable.* The representation convergence of interactions has long been observed (Li & Zhang, 2023a). The method to identify convergent (*i.e.,* generalizable) interactions will be introduced later in Equation (5).

**(2)** *A DNN's overfitting* is measured by the gap between training and testing losses, following standard practice (Zhang et al., 2019).

Thus, our central question is: **How does the representation convergence of interactions relate to a DNN's overfitting?** To answer this, we examine the following two propositions.

**Proposition 1, descriptive study:** *When several DNNs are trained on the same task, the representation convergence of interactions across these DNNs increases as long as the training-testing loss gap has not yet exhibited signs of overfitting. Once the loss gap begins to widen, the representation convergence correspondingly declines.*

**Proposition 2, interventional study:** *Proactively eliminating non-convergent interactions during training directly enhances the convergence of interaction representations, which in turn substantially reduces the training-testing gap of the cross-entropy loss.*

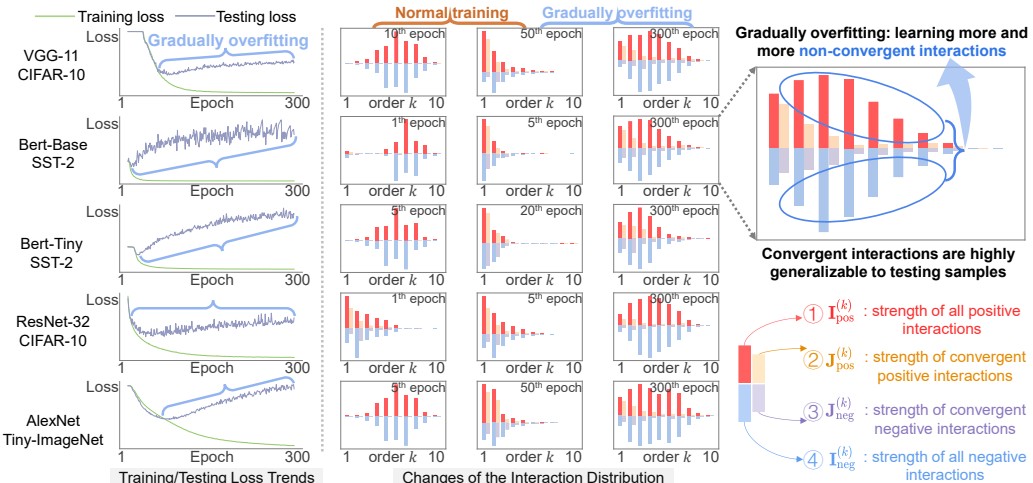

Figure 2: The starting point of the DNN's overfitting is characterized by the gradual learning of numerous non-convergent interactions[4]. These non-convergent interactions are often of high orders and mutually offsetting. In contrast, the convergent interactions are **highly generalizable to testing samples**, because the baseline DNNs in this figure are trained on the testing set. We identify the time point for the trend of overfitting as the point where the loss gap begins to widen.

**Successfully validating these propositions would establish a strong connection between the representation convergence of interactions and a DNN's overfitting.** Specifically, as illustrated in Figure 2, during the training process, a decrease in interaction convergence temporally aligns with the onset of a DNN's overfitting. Moreover, proactively eliminating non-convergent interactions (*i.e.,* enhancing interaction convergence) can consistently reduce the degree of a DNN's overfitting. These propositions, from both descriptive and interventional perspectives, demonstrate a causal relationship between the representation convergence of interactions and a DNN's overfitting.

# 3 VALIDATION OF THE TWO PROPOSITIONS

## 3.1 VALIDATION OF PROPOSITION 1

**Quantifying the representation convergence of interactions between DNNs.** Given a target DNN $v$, we train an additional baseline DNN $v'$ for the same task. Intuitively, if both $v$ and $v'$ independently encode the same interaction and it has a similar effect, this interaction is considered a *convergent interaction*. Specifically, for each AND interaction $S$, we follow (Chen et al., 2024) and suppose a convergent interaction $S$ to be simultaneously extracted a salient interaction by both the target DNN $v$ and the baseline DNN $v'$ (*s.t.* $S \in \Omega_{\text{sparse}}^{\text{AND}}$ and $S \in \Omega_{\text{sparse},v'}^{\text{AND}}$) and make a similar effect to classify the input sample $\boldsymbol{x}$ (*s.t.* $I_S^{\text{AND}} \cdot I_{S,v'}^{\text{AND}} > 0$), where $\Omega_{\text{sparse}}^{\text{AND}}$ and $\Omega_{\text{sparse},v'}^{\text{AND}}$ denote the sets of salient AND interactions extracted by the target DNN $v$ and the baseline DNN $v'$, respectively. Thus, the binary metric of identifying convergent interactions is given as follows.

$$\mathcal{G}_S^{\text{type}} \stackrel{\text{def}}{=} \mathbb{1}(S \in \Omega_{\text{sparse}}^{\text{type}}) \cdot \mathbb{1}(S \in \Omega_{\text{sparse},v'}^{\text{type}}) \cdot \mathbb{1}(I_S^{\text{type}} \cdot I_{S,v'}^{\text{type}} > 0), \quad \text{type} \in \{\text{AND}, \text{OR}\}, \quad (5)$$

where $\mathbb{1}(\cdot) \in \{0, 1\}$ is a trigger function that returns 1 if the given condition is satisfied. The representation consistency of OR interactions is defined in the same manner.

Then, we use the following four metrics to quantify the strength of convergent interactions over different orders, including (1) the strength of all positive interactions of the $k$-th order $\mathbf{I}_{\text{pos}}^{(k)}$, (2) the strength of all negative interactions of the $k$-th order $\mathbf{I}_{\text{neg}}^{(k)}$, (3) the strength of all convergent positive interactions $\mathbf{J}_{\text{pos}}^{(k)}$, and (4) the strength of all convergent negative interactions $\mathbf{J}_{\text{neg}}^{(k)}$.

$$\mathbf{I}_{\text{pos}}^{(k)} = \sum_{\text{type} \in \{\text{AND, OR}\}} \sum_{S \in \Omega_{\text{sparse}}^{\text{type}} : |S| = k} \max(I_S^{\text{type}}, 0), \quad \mathbf{J}_{\text{pos}}^{(k)} = \sum_{\text{type} \in \{\text{AND, OR}\}} \sum_{S \in \Omega_{\text{sparse}}^{\text{type}} : |S| = k} \max(I_S^{\text{type}} \cdot \mathcal{G}_S^{\text{type}}, 0), \quad (6)$$

$$\mathbf{I}_{\text{neg}}^{(k)} = \sum_{\text{type} \in \{\text{AND, OR}\}} \sum_{S \in \Omega_{\text{sparse}}^{\text{type}} : |S| = k} \min(I_S^{\text{type}}, 0), \quad \mathbf{J}_{\text{neg}}^{(k)} = \sum_{\text{type} \in \{\text{AND, OR}\}} \sum_{S \in \Omega_{\text{sparse}}^{\text{type}} : |S| = k} \min(I_S^{\text{type}} \cdot \mathcal{G}_S^{\text{type}}, 0). \quad (7)$$

---

[4]The distribution of interactions is computed by averaging the distribution across different samples.

**Experimental validation of Proposition 1.** To evaluate whether the representation convergence of interactions and the overfitting degree of a DNN are strongly negatively correlated, as proposed by Proposition 1, we tracked the evolution of the four aforementioned metrics throughout the entire training process, alongside the change of the training-testing loss gap. Specifically, we trained VGG-11 (Simonyan, 2014) and ResNet-32 (He et al., 2016) on the CIFAR-10 dataset (Krizhevsky et al., 2009), AlexNet (Krizhevsky et al., 2012) on the Tiny-ImageNet dataset (mnmoustafa, 2017), and both Bert-Tiny and Bert-Base (Devlin, 2018) models on the SST-2 dataset (Socher et al., 2013). For image data, we followed (Ren et al., 2024b) to select ten patches from each image as input variables; for natural language data, we treated the embedding of each word as an input variable. Please see Appendix J.3 for details. In addition, we adopted the threshold $\tau = 0.02 \cdot \mathbb{E}_{\boldsymbol{x}}[|v(\boldsymbol{x}) - v(\boldsymbol{x}_\emptyset)|]$ as proposed in (Ren et al., 2024b).

Figure 2 shows that the trend of all interactions (*i.e.,* $\mathbf{I}_{\text{pos}}^{(k)}$ and $\mathbf{I}_{\text{neg}}^{(k)}$) was closely aligned with the trend of convergent interactions (*i.e.,* $\mathbf{J}_{\text{pos}}^{(k)}$ and $\mathbf{J}_{\text{neg}}^{(k)}$) during both the normal training phase (when the training-testing loss gap is relatively small) and the overfitting phase (when the training-testing loss gap begins to widen). *In particular, because we trained the baseline DNN on testing samples in this experiment, such specific convergent interactions in the target DNN can be regarded **generalizable to testing samples**[5]. We found that when the loss gap was relatively small, the DNN increasingly encoded convergent/generalizable interactions. Once the loss gap began to widen, the representation convergence (the interactions' generalizability to testing samples) gradually declined, as the network increasingly encoded non-convergent interactions. *These results indicated that the representation convergence of interactions and the overfitting degree of a DNN were strongly negatively correlated, thereby validating Proposition 1.*

## 3.2 VALIDATION OF PROPOSITION 2

To validate Proposition 2, *i.e.,* whether actively eliminating non-convergent interactions can effectively mitigate a DNN's overfitting, we employ an interventional approach that removes the non-convergent interactions encoded by the DNN, and we then examine whether the training-testing cross-entropy loss gap decreases. The validation of Proposition 2 provides strong evidence of a close connection between the representation convergence of interactions and a DNN's overfitting.

**Implementation of eliminating non-convergent interactions.** Given a target DNN $v$ and a training sample $\boldsymbol{x}$, according to Equation (5), the elimination of the non-convergent interactions encoded by the target DNN $v$ can be characterized by the following objective.

$$\min \mathbb{E}_{\boldsymbol{x}}\left[ \sum_{S \subseteq N} \sum_{\text{type} \in \{\text{AND,OR}\}} \left(1 - \mathcal{G}_S^{\text{type}}(\boldsymbol{x})\right) \cdot |I_S^{\text{type}}(\boldsymbol{x})| \right], \tag{8}$$

where $\left(1 - \mathcal{G}_S^{\text{type}}(\boldsymbol{x})\right) \cdot |I_S^{\text{type}}(\boldsymbol{x})|$ denotes the strength of non-convergent interactions in the DNN $v$.

The above objective directly eliminates non-convergent interactions from $v$, measured *w.r.t.* the baseline DNN $v'$ (see Equation (5)), but it is difficult to optimize. To address this, we introduce the following relaxation. First, instead of using a fixed baseline DNN, we jointly train two DNNs, $v$ and $v'$, as surrogates for the target DNN and the baseline DNN, respectively, and we encourage them to encode similar interactions, *i.e.,* $\forall S \subseteq N, \forall \text{type} \in \{\text{AND, OR}\}, I_S^{\text{type}}(\boldsymbol{x}) \sim I_{S,v'}^{\text{type}}(\boldsymbol{x})$, where $I_S^{\text{type}}(\boldsymbol{x})$ and $I_{S,v'}^{\text{type}}(\boldsymbol{x})$ denote the AND/OR interactions extracted by the DNN $v$ and the DNN $v'$, respectively. Besides, instead of directly penalizing $\mathbb{E}_{S \subseteq N} \sum_{\text{type} \in \{\text{AND,OR}\}} [I_S^{\text{type}}(\boldsymbol{x}) - I_{S,v'}^{\text{type}}(\boldsymbol{x})]^2$, which incurs computational cost of $\mathcal{O}(2^{|N|})$, we propose to train both $v$ and $v'$ with the following loss function $\hat{\mathcal{L}}$. **Theoretical analysis in Appendix C demonstrates that minimizing $\hat{\mathcal{L}}$ is an approximate-yet-efficient method to encourage the DNN $v$ and the DNN $v'$ to encode similar interactions.**

$$\hat{\mathcal{L}}(v, v') = \mathbb{E}_{(\boldsymbol{x}, \boldsymbol{y}) \in D} \left[ \lambda \cdot \mathbb{E}_{S_1, S_2, S_3} \left[ (\mathcal{I}_v(S_1, S_2 | S_3, \boldsymbol{x}) - \mathcal{I}_{v'}(S_1, S_2 | S_3, \boldsymbol{x}))^2 \right] + (1 - \lambda) \cdot \mathcal{L}_{\text{CE}}(\boldsymbol{x}, \boldsymbol{y}) \right], \tag{9}$$

$$\text{s.t.} \begin{cases} \mathcal{I}_v(S_1, S_2 | S_3, \boldsymbol{x}) = v(\boldsymbol{x}_{S_1 \cup S_2 \cup S_3}) - v(\boldsymbol{x}_{S_1 \cup S_3}) - v(\boldsymbol{x}_{S_2 \cup S_3}) + v(\boldsymbol{x}_{S_3}) \\ \mathcal{I}_{v'}(S_1, S_2 | S_3, \boldsymbol{x}) = v'(\boldsymbol{x}_{S_1 \cup S_2 \cup S_3}) - v'(\boldsymbol{x}_{S_1 \cup S_3}) - v'(\boldsymbol{x}_{S_2 \cup S_3}) + v'(\boldsymbol{x}_{S_3}) \end{cases}, \tag{10}$$

where $S_1, S_2, S_3 \subseteq N$ are three disjoint subsets of the input variables, subject to $S_1 \cap S_2 = S_1 \cap S_3 = S_2 \cap S_3 = \emptyset$. $\mathcal{I}_v(S_1, S_2 | S_3, \boldsymbol{x})$ and $\mathcal{I}_{v'}(S_1, S_2 | S_3, \boldsymbol{x})$ denote the simplified interactions between the

---

[5]The baseline DNN also uses these interactions to classify testing samples.

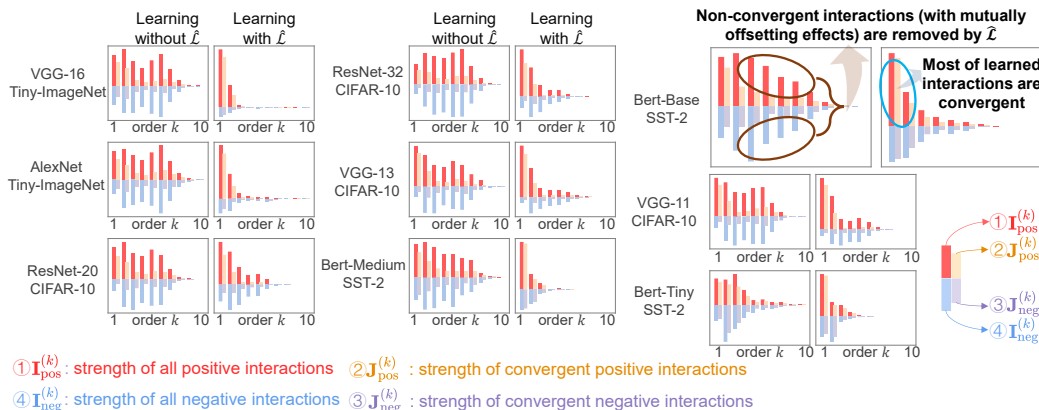

Figure 3: Our method enhances representation convergence of interactions by eliminating non-convergent interactions while preserving convergent interactions[4].

variables in $S_1$ and $S_2$, conditioned on $S_3$, as extracted by the DNNs $v$ and $v'$, respectively. Further details are provided in Appendix C. $\mathcal{L}_{\mathrm{CE}}$ denotes the cross-entropy loss. $(\boldsymbol{x}, \boldsymbol{y})$ denotes a training sample and its corresponding label from the dataset $\mathcal{D}$. $\lambda \in (0, 1)$ is a scalar weight.

**Experiment 1: Verifying the elimination of non-convergent interactions.** We conducted experiments to verify whether the proposed loss function $\hat{\mathcal{L}}$ effectively eliminated non-convergent interactions in Equation (8). Specifically, we trained[6] VGG-11/13 and ResNet-20/32 on the CIFAR-10 dataset, trained[6] AlexNet and VGG-16 on the Tiny-ImageNet dataset, and trained[6] Bert-Tiny, Bert-Medium, and Bert-Base on the SST-2 dataset. Please see Appendix J for detailed settings.

As shown in Figure 3, compared with training the DNN using only cross-entropy loss, *the proposed loss function $\hat{\mathcal{L}}$ successfully eliminated non-convergent interactions and increased the proportion of convergent ones among all interactions encoded by the target DNN.*

**Experiment 2: Verifying whether eliminating non-convergent interactions effectively reduces the training-testing cross-entropy loss gap.** We conducted experiments to compare the training-testing cross-entropy loss gap between DNNs trained by setting different $\lambda$ values in Equation (9) to eliminate different amounts of non-convergent interactions. Specifically, we trained[6] ResNet-20, VGG-11, and VGG-13 on the CIFAR-10 dataset, and Bert-Tiny on the SST-2 dataset. Please see Appendix J for detailed settings.

As shown in Figure 4(b), *higher $\lambda$ values led to more removal of non-convergent interactions, which in turn significantly reduced the training-testing cross-entropy loss gap. These results indicated that eliminating non-convergent interactions effectively mitigated the degree of a DNN's overfitting, thereby verifying Proposition 2.*

**Experiment 3: Verifying the strong causal relationship between non-convergent interactions and a DNN's overfitting by intermittently eliminating non-convergent interactions.** If this causal relation holds, the degree of overfitting should vary intermittently with changes in non-convergent interactions. In this way, we conducted an experiment to examine the effect of intermittently eliminating non-convergent interactions in the following three phases. (1) Phase 1: non-convergent interactions were eliminated by setting $\lambda = 0.8$ in the 1st-500th epochs. (2) Phase 2: elimination was suspended by setting $\lambda = 0$ in the 501st-1000th epochs. (3) Phase 3: elimination was resumed by resetting $\lambda = 0.8$ in the 1001st-1500th epochs. We conducted experiments to examine whether the loss gap increased when elimination was suspended and continued to decrease when elimination resumed. Specifically, we trained[6] ResNet-20, VGG-11, and VGG-13 on the CIFAR-10 dataset, as well as Bert-Tiny on the SST-2 dataset for 300 epochs.

As shown in Figure 4(a), whenever the interaction convergence loss was applied (*i.e.,* setting $\lambda = 0.8$), the training-testing cross-entropy loss gap continuously decreased until convergence.

---

[6]For each network architecture, we trained two DNNs with differently initialized parameters, which served as the target DNN $v$ and the baseline DNN $v'$, respectively.

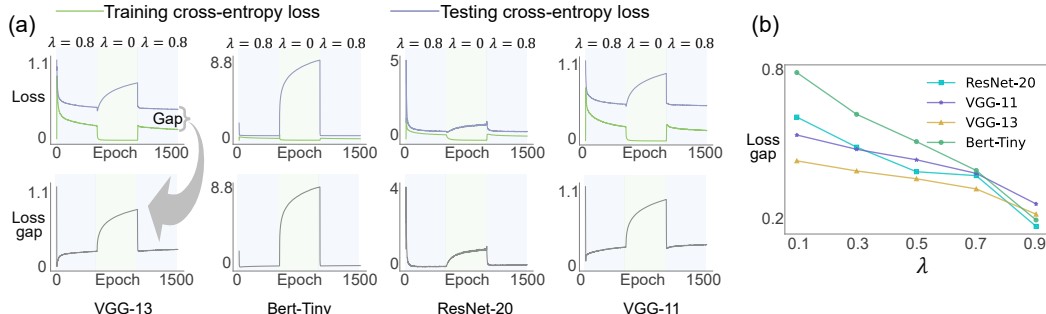

Figure 4: (a) The cross-entropy loss curves on the training and testing sets. We manipulated the training process by temporarily disabling the elimination of non-convergent interactions between the 501st and 1000th epochs by setting $\lambda = 0$. We found that eliminating non-convergent interactions effectively controlled the testing-training cross-entropy loss gap, preventing it from continuously widening. However, whenever the elimination was suspended, the loss gap gradually widened. (b) The cross-entropy loss gap decreased as the $\lambda$ value increased.

Conversely, whenever the interaction convergence loss was removed (*i.e.,* setting $\lambda = 0$), the testing cross-entropy loss increased while the training cross-entropy loss approached zero. *These results further indicated that the elimination of non-convergent interactions effectively mitigated the degree of a DNN's overfitting, thereby verifying Proposition 2.*

### 3.3 IMPROVING INTERACTION QUALITY IS THE KEY TO MITIGATING DNN OVERFITTING

In this section, we explore why eliminating non-convergent interactions mitigates a DNN's overfitting. We find that the root reason is that this elimination improves the representational quality of the remaining interactions. In fact, the quality of interactions is a key factor in determining the degree of a DNN's overfitting.

In the field of symbolic generalization, prior studies have provided some empirical conclusions. Two typical findings on interactions are as follows.
(1) High-order interactions tend to generalize less compared to low-order ones (Zhou et al., 2024).
(2) Mutually offsetting interactions, where the strength of positive interaction effects was similar to that of negative interaction effects, typically correspond to non-generalizable noise induced by overfitting (Cheng et al., 2025).

The above evaluation metrics of interaction complexity/order enable us to clarify **the root reason why eliminating non-convergent interactions mitigates a DNN's overfitting: it effectively removes (1) high-order interactions and (2) mutually offsetting interactions.** Specifically, as shown in Figure 3, the proposed loss function $\hat{\mathcal{L}}$ eliminated many high-order interactions (*e.g.,* the 6th–8th orders) while preserving low-order ones (*i.e.,* mostly limited to the 4th order or below). Meanwhile, $\hat{\mathcal{L}}$ reduced the prevalence of mutually offsetting interactions. *The preserved low-order, non-offsetting interactions typically corresponded to high-quality patterns that generalized well to unseen testing data.*

**Boosting the ratio of interactions that generalize to testing samples**. In addition to promoting convergence across DNNs, we find that eliminating non-convergent interactions also enhances those interactions' generalizability to testing samples, which represents a more widely recognized form of generalizability. Although difficult to model directly, this generalizability can be approximated by the ratio of interactions in the target DNN that converge to those in a **baseline DNN trained on testing samples**. Such specific convergent interactions can be viewed as primitive patterns inherent in the testing set. **Appendix F illustrates the contribution of our proposed loss function $\hat{\mathcal{L}}$ to symbolic generalization. Specifically, Figure 7 shows that eliminating non-convergent interactions successfully forced the DNN to encode more interactions that generalize to testing samples.**

### 3.4 TOWARDS THE FIRST-PRINCIPLES EXPLANATION OF A DNN'S OVERFITTING.

Although we cannot yet claim that the current findings constitute a true first-principles explanation, theoretical rigor compels us to explore a follow-up question: **Is the elimination of non-convergent**

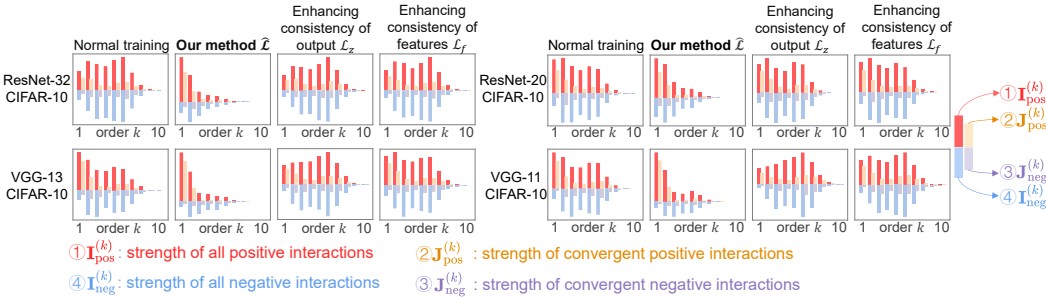

Figure 5: Comparing distributions[4] of interactions learned by different methods. Only our method effectively eliminates non-convergent interactions from DNNs.

interactions the only way to mitigate overfitting? In other words, can other forms of feature consistency across DNNs also mitigate overfitting? To answer this, we further explore the effectiveness of two widely used loss functions in knowledge distillation (Hinton et al., 2015; Romero et al., 2014). The first loss function penalizes the squared error between the output $\mathbf{z}_1(\boldsymbol{x})$ and $\mathbf{z}_2(\boldsymbol{x})$ before the softmax layer of two DNNs $v$ and $v'$, respectively, as follows.

$$\mathcal{L}_z = \mathbb{E}_{(\boldsymbol{x},\boldsymbol{y}) \in D} \left[ \lambda \cdot \|\mathbf{z}_1(\boldsymbol{x}) - \mathbf{z}_2(\boldsymbol{x})\|_2^2 + (1 - \lambda) \cdot \mathcal{L}_{\text{CE}}(\boldsymbol{x}, \boldsymbol{y}) \right], \tag{11}$$

where $\|\cdot\|_2^2$ denotes the squared $\ell_2$-norm. The second loss function penalizes the squared error between intermediate-layer features $\mathbf{f}_1(\boldsymbol{x})$ and $\mathbf{f}_2(\boldsymbol{x})$ of two DNNs $v$ and $v'$, as follows:

$$\mathcal{L}_f = \mathbb{E}_{(\boldsymbol{x},\boldsymbol{y}) \in D} \left[ \lambda \cdot \|\mathbf{f}_1(\boldsymbol{x}) - \mathbf{f}_2(\boldsymbol{x})\|_2^2 + (1 - \lambda) \cdot \mathcal{L}_{\text{CE}}(\boldsymbol{x}, \boldsymbol{y}) \right]. \tag{12}$$

We conducted experiments to compare DNNs trained using our proposed loss function $\hat{\mathcal{L}}$ with DNNs trained on the above two competing loss functions. Specifically, We trained VGG-11, VGG-13, ResNet-20, and ResNet-32 on the CIFAR-10 dataset. Please see Appendix J for detailed settings.

Figure 5 compares the distributions of interactions between DNNs trained with our proposed loss function $\hat{\mathcal{L}}$ and those trained with the two competing loss functions. We observed that only our method effectively eliminated non-convergent interactions, whereas the competing methods failed to do so. Furthermore, Figure 9 in the Appendix shows that only our method successfully reduced the training-testing cross-entropy loss, while the competing methods failed to achieve such a reduction. These results indicated that **the elimination of non-convergent interactions was the only way to mitigate overfitting, while other forms of feature consistency did not have this effect.**

## 4 CONCLUSION AND DISCUSSION

In this paper, we have proposed and validated two propositions that establish a rigorous relationship between a DNN's overfitting and the representation convergence of interactions. Specifically, Proposition 1 demonstrates from a descriptive perspective that the representation convergence of interactions and the overfitting degree of a DNN are strongly negatively correlated, while Proposition 2 shows from an interventional perspective that enhancing the representation convergence of interactions successfully mitigates overfitting. Experimental validation of the two propositions confirms a strong causal relationship between a DNN's overfitting and the representation convergence of interactions from dual perspectives. Furthermore, we also find that the eliminated non-convergent interactions mainly represent non-generalizable patterns (most are high-order and mutually offsetting interaction effects), which explains the mitigation of the DNN's overfitting.

**Practical value.** This work is the first to rigorously link a DNN's generalization power to the generalizability of its encoded interactions. Therefore, our explanation theory offers a principled approach to determining the optimal early stopping point during training, by objectively characterizing how the generalizability of interactions evolves over time. Accordingly, by monitoring this evolution throughout the training process, one can stop training when interaction generalizability reaches its peak, providing a new perspective grounded in the fine-grained inference patterns learned by the DNN. Moreover, we also find that eliminating non-convergent interactions can slightly enhance overall performance (testing accuracy) in models. Please see Appendix E for the detailed results.

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

## A  RELATED WORK

Understanding and quantifying the generalization power of deep neural networks (DNNs) remains a central problem in deep learning. Prior studies have largely approached this issue from two main perspectives: (1) evaluating the loss gap between training and testing set (Neyshabur et al., 2017; Bousquet et al., 2020; Deng et al., 2021; Haghifam et al., 2021; 2020), and (2) characterizing the smoothness of the loss landscape (Li et al., 2018; Keskar et al., 2016; Kwon et al., 2021). Beyond these perspectives, further efforts have explored the generalization power of DNNs in high-dimensional feature space (Petrini et al., 2022; Nikolikj et al., 2024; Boopathy et al., 2023).

However, recent studies on the generalization ability of AI models have shifted toward more refined and nuanced perspectives. Within this shift, the emerging line of research on **symbolic generalization** (Ren et al., 2023; Li & Zhang, 2023a; Ren et al., 2024a;b; Zhou et al., 2024) has attracted growing attention. This theory offers a novel framework for faithfully explaining the inference patterns encoded by DNNs. Specifically, it has been discovered (Ren et al., 2023) and proven (Ren et al., 2024a) that a DNN's output scores can be accurately mimicked by an AND-OR logical model, which contains a few AND-OR interactions. Furthermore, Chen et al. (2024) proposed an algorithm to extract shared interactions. As a novel mathematical tool, interaction has also revealed the underlying mechanisms of traditionally empirical algorithms. For example, Wang et al. (2020) found that all twelve adversarial transferability methods implicitly decreased interactions between adversarial perturbations. Ren et al. (2021) revealed that fourteen popular attribution methods could be explained as reallocation of interaction effects.

Furthermore, symbolic generalization theory also provides a new perspective to explain the generalization power of a DNN. Li & Zhang (2023a) have discovered that many interactions encoded by a DNN can generalize across different samples. In particular, Zhou et al. (2024) have found that simple interactions have higher generalization power than complex interactions. Moreover, Zhang et al. (2024) discovered and Ren et al. (2024b) proved a two-phase dynamics of learning interactions during the training process of a DNN, and they found that numerous complex interactions emerge in the overfitting phase of the DNN.

However, prior descriptive studies have not established a rigorous connection between the overall generalization of DNNs and the generalizability of their interactions, *i.e.,* a gap that remains one of the central challenges in symbolic generalization (Zhou et al., 2024; Ren et al., 2024b; Chen et al., 2024). To tackle this issue, we propose and validate two propositions, which jointly provide complementary evidence that overfitting in a DNN is closely tied to the convergence of its symbolic interactions. Validating these propositions from both descriptive and interventional perspectives further reinforces the theoretical foundation of symbolic generalization.

**Comparison with mechanistic interpretability.** In contrast to prior mechanistic interpretability studies (Olah, 2022; Nanda et al., 2023; Zhou et al., 2018; Meng et al., 2022; Liu et al., 2023b; Lieberum et al., 2023; Wang et al., 2022), which explain representations of explicit neurons inside the DNN (*e.g.,* neuron circuits or local causal structures), symbolic generalization explains the inference patterns of a DNN by identifying a small set of salient interactions between input variables in a post-hoc manner. By design, the symbolic generalization theory treats the entire DNN as a black-box model, without attempting to interpret the roles of specific neurons.

## B  THE USE OF LARGE LANGUAGE MODELS (LLMS)

In this work, we employed a large language model (LLM) solely for the purpose of improving the clarity and readability of the manuscript by refining the language and phrasing. All aspects of the research were conducted independently by the authors. The LLM did not contribute to research ideation or substantive content development.

## C  THEORETICAL JUSTIFICATION: WHY THE PROPOSED LOSS ENCOURAGES TWO DNNs TO ENCODE SIMILAR INTERACTIONS

The simplified interactions and the new loss function based on them are defined as follows.

$$\hat{\mathcal{L}}(v, v') = \mathbb{E}_{(\boldsymbol{x}, \boldsymbol{y}) \in D} \left[ \lambda \cdot \mathbb{E}_{S_1, S_2, S_3} \left[ (\mathcal{I}_v(S_1, S_2 | S_3, \boldsymbol{x}) - \mathcal{I}_{v'}(S_1, S_2 | S_3, \boldsymbol{x}))^2 \right] + (1 - \lambda) \cdot \mathcal{L}_{\text{CE}}(\boldsymbol{x}, \boldsymbol{y}) \right], \quad (13)$$

$$\text{s.t.} \begin{cases} \mathcal{I}_v(S_1, S_2 | S_3, \boldsymbol{x}) = v(\boldsymbol{x}_{S_1 \cup S_2 \cup S_3}) - v(\boldsymbol{x}_{S_1 \cup S_3}) - v(\boldsymbol{x}_{S_2 \cup S_3}) + v(\boldsymbol{x}_{S_3}) \\ \mathcal{I}_{v'}(S_1, S_2 | S_3, \boldsymbol{x}) = v'(\boldsymbol{x}_{S_1 \cup S_2 \cup S_3}) - v'(\boldsymbol{x}_{S_1 \cup S_3}) - v'(\boldsymbol{x}_{S_2 \cup S_3}) + v'(\boldsymbol{x}_{S_3}) \end{cases}, \quad (14)$$

where $S_1, S_2, S_3 \subseteq N$ are three disjoint subsets of the input variables in $N$, subject to $S_1 \cap S_2 = S_1 \cap S_3 = S_2 \cap S_3 = \emptyset$. $\mathcal{I}_v(S_1, S_2 | S_3, \boldsymbol{x})$ represents the simplified interaction (*i.e.,* non-linear relationship) between input variables in $S_1$ and those in $S_2$, extracted by the DNN $v$ from the given input sample $\boldsymbol{x}$, conditioned on the variables in $S_3$. Similarly, $\mathcal{I}_{v'}(S_1, S_2 | S_3, \boldsymbol{x})$ denotes the simplified interaction extracted by the DNN $v'$. $\mathbb{E}_{S_1, S_2, S_3} \left[ (\mathcal{I}_v(S_1, S_2 | S_3, \boldsymbol{x}) - \mathcal{I}_{v'}(S_1, S_2 | S_3, \boldsymbol{x}))^2 \right]$ denotes interaction convergence loss. $\mathcal{L}_{\text{CE}}$ denotes the cross-entropy loss.

To justify why the proposed loss function $\hat{\mathcal{L}}$ encourages two DNNs $v$ and $v'$ to encode similar interactions, we analyze the structure of the simplified interaction term $\mathcal{I}_v(S_1, S_2 | S_3, \boldsymbol{x})$, and how the interaction consistency loss,

$$\mathbb{E}_{S_1, S_2, S_3} \left[ (\mathcal{I}_v(S_1, S_2 | S_3, \boldsymbol{x}) - \mathcal{I}_{v'}(S_1, S_2 | S_3, \boldsymbol{x}))^2 \right],$$

encourages the two DNNs $v$ and $v'$ to encode similar interactions.

### C.1  DECOMPOSITION OF SIMPLIFIED INTERACTIONS

For a fixed input $\boldsymbol{x}$, we consider the simplified interaction between variable sets $S_1$ and $S_2$ given $S_3$, defined as:

$$\mathcal{I}_v(S_1, S_2 | S_3, \boldsymbol{x}) = v(\boldsymbol{x}_{S_1 \cup S_2 \cup S_3}) - v(\boldsymbol{x}_{S_1 \cup S_3}) - v(\boldsymbol{x}_{S_2 \cup S_3}) + v(\boldsymbol{x}_{S_3}). \quad (15)$$

According to the proof of the universal-matching property in Appendix D, we have:

$$v(\boldsymbol{x}_S) = u_S^{\text{AND}} + u_S^{\text{OR}} = v(\boldsymbol{x}_\emptyset) + \sum_{\emptyset \neq T \subseteq S} I_T^{\text{AND}} + \sum_{T: T \cap S \neq \emptyset} I_T^{\text{OR}}. \quad (16)$$

Focusing on the AND component (the process for the OR interaction follows the same logic), we define $v(\boldsymbol{x}_S)$ as:

$$v(\boldsymbol{x}_S) := \sum_{\emptyset \neq T \subseteq S} I_T^{\text{AND}}. \quad (17)$$

Let us explicitly write out the expressions for several specific subsets:

$$v(\boldsymbol{x}_{S_1 \cup S_2 \cup S_3}) = I_{S_1}^{\text{AND}} + I_{S_2}^{\text{AND}} + I_{S_3}^{\text{AND}} + I_{S_1 \cup S_2}^{\text{AND}} + I_{S_1 \cup S_3}^{\text{AND}} + I_{S_2 \cup S_3}^{\text{AND}} \quad (18)$$

$$v(\boldsymbol{x}_{S_1 \cup S_2}) = I_{S_1}^{\text{AND}} + I_{S_2}^{\text{AND}} + I_{S_1 \cup S_2}^{\text{AND}} \quad (19)$$

$$v(\boldsymbol{x}_{S_1 \cup S_3}) = I_{S_1}^{\text{AND}} + I_{S_3}^{\text{AND}} + I_{S_1 \cup S_3}^{\text{AND}} \quad (20)$$

$$v(\boldsymbol{x}_{S_2 \cup S_3}) = I_{S_2}^{\text{AND}} + I_{S_3}^{\text{AND}} + I_{S_2 \cup S_3}^{\text{AND}} \quad (21)$$

$$v(\boldsymbol{x}_{S_3}) = I_{S_3}^{\text{AND}} \quad (22)$$

In this way, we have

$$
\begin{aligned}
\mathcal{I}_v(S_1, S_2 | S_3, \boldsymbol{x}) &= v(\boldsymbol{x}_{S_1 \cup S_2 \cup S_3}) - v(\boldsymbol{x}_{S_1 \cup S_3}) - v(\boldsymbol{x}_{S_2 \cup S_3}) + v(\boldsymbol{x}_{S_3}) \\
&= \left( I_{S_1}^{\mathrm{AND}} + I_{S_2}^{\mathrm{AND}} + I_{S_3}^{\mathrm{AND}} + I_{S_1 \cup S_2}^{\mathrm{AND}} + I_{S_1 \cup S_3}^{\mathrm{AND}} + I_{S_2 \cup S_3}^{\mathrm{AND}} \right) \\
&\quad - \left( I_{S_1}^{\mathrm{AND}} + I_{S_3}^{\mathrm{AND}} + I_{S_1 \cup S_3}^{\mathrm{AND}} \right) \\
&\quad - \left( I_{S_2}^{\mathrm{AND}} + I_{S_3}^{\mathrm{AND}} + I_{S_2 \cup S_3}^{\mathrm{AND}} \right) \\
&\quad + I_{S_3}^{\mathrm{AND}} \\
&= I_{S_1 \cup S_2}^{\mathrm{AND}}
\end{aligned}
\tag{23}
$$

This result shows that $\mathcal{I}_v(S_1, S_2 | S_3, \boldsymbol{x})$ represents the simplified interaction (*i.e.,* non-linear relationship) between input variables in $S_1$ and those in $S_2$, extracted by $v$ from the given input sample $\boldsymbol{x}$, conditioned on the variables in $S_3$. Similarly, $\mathcal{I}_{v'}(S_1, S_2 | S_3, \boldsymbol{x})$ denotes the simplified interaction extracted by the DNN $v'$.

### C.2 EFFECT OF THE INTERACTION CONSISTENCY LOSS

The interaction consistency loss penalizes the squared difference between the simplified interactions extracted by $v$ and $v'$ across many randomly sampled disjoint subsets $(S_1, S_2, S_3)$. Concretely,

$$
\mathbb{E}_{S_1, S_2, S_3} \left[ \left( \mathcal{I}_v(S_1, S_2 | S_3, \boldsymbol{x}) - \mathcal{I}_{v'}(S_1, S_2 | S_3, \boldsymbol{x}) \right)^2 \right]
$$

acts as a regularizer that encourages the two networks to encode similar interactions.

Therefore, by minimizing the interaction consistency loss over numerous tuples $(S_1, S_2, S_3)$, the proposed objective $\hat{\mathcal{L}}$ encourages the two DNNs to encode similar interactions between input variables in $S_1$ and those in $S_2$, conditioned on the variables in $S_3$

### C.3 EFFICIENCY OF THE INTERACTION CONSISTENCY LOSS

The loss based on the simplified interaction can be computed with significantly lower computational cost. It is because in real-world applications, it is not necessary to exhaustively enumerate all possible divisions of the three subsets $(S_1, S_2, S_3)$, subject to $S_1, S_2, S_3 \subseteq N, S_1 \cap S_2 = S_1 \cap S_3 = S_2 \cap S_3 = \emptyset$. Instead, we can simply sample a random triplet of $(S_1, S_2, S_3)$ to compute the loss $\hat{\mathcal{L}}$ on the input sample $\boldsymbol{x}$ in each training batch. This significantly improves computational efficiency.

## D  PROOF OF THEOREM 3

*Proof.* **(1) Universal matching theorem of AND interactions.**

We will prove that output component $u_S^{\mathrm{AND}}$ on all $2^n$ masked samples $\{\boldsymbol{x}_S : S \subseteq N\}$ could be universally explained by the all interactions in $S \subseteq N$, *i.e.,* $\forall \emptyset \neq S \subseteq N, u_S^{\mathrm{AND}} = \sum_{\emptyset \neq T \subseteq S} I_T^{\mathrm{AND}} + v(\boldsymbol{x}_\emptyset)$. In particular, we define $v_\emptyset^{\mathrm{AND}} = v(\boldsymbol{x}_\emptyset)$ (*i.e.,* we attribute output on an empty sample to AND interactions).

Specifically, the AND interaction is defined as $I_T^{\mathrm{AND}} = \sum_{L \subseteq T} (-1)^{|T|-|L|} u_L^{\mathrm{AND}}$. To compute the sum of AND interactions $\sum_{\emptyset \neq T \subseteq S} I_T^{\mathrm{AND}} = \sum_{\emptyset \neq T \subseteq S} \sum_{L \subseteq T} (-1)^{|T|-|L|} u_L^{\mathrm{AND}}$, we first exchange the order of summation of the set $L \subseteq T \subseteq S$ and the set $T \supseteq L$. That is, we compute all linear combinations of all sets $T$ containing $L$ with respect to the model outputs $u_L^{\mathrm{AND}}$ given a set of input variables $L$, *i.e.,* $\sum_{T: L \subseteq T \subseteq S} (-1)^{|T|-|L|} u_L^{\mathrm{AND}}$. Then, we compute all summations over the set $L \subseteq S$.

In this way, we can compute them separately for different cases of $L \subseteq T \subseteq S$. In the following, we consider the cases (1) $L = S = T$, and (2) $L \subseteq T \subseteq S, L \neq S$, respectively.

(1) When $L = S = T$, the linear combination of all subsets $T$ containing $L$ with respect to the model output $u_L^{\mathrm{AND}}$ is $(-1)^{|S|-|S|} u_L^{\mathrm{AND}} = u_L^{\mathrm{AND}}$.

(2) When $L \subseteq T \subseteq S, L \neq S$, the linear combination of all subsets $T$ containing $L$ with respect to the model output $u_L^{\mathrm{AND}}$ is $\sum_{T: L \subseteq T \subseteq S} (-1)^{|T|-|L|} u_L^{\mathrm{AND}}$. For all sets $T : S \supseteq T \supseteq L$, let us consider

the linear combinations of all sets $T$ with number $|T|$ for the model output $u_L^{\text{AND}}$, respectively. Let $m := |T| - |L|$, $(0 \le m \le |S| - |L|)$, then there are a total of $C_{|S|-|L|}^m$ combinations of all sets $T$ of order $|T|$. Thus, given $L$, accumulating the model outputs $u_L^{\text{AND}}$ corresponding to all $T \supseteq L$, then $\sum_{T:L \subseteq T \subseteq S}(-1)^{|T|-|L|}u_L^{\text{AND}} = u_L^{\text{AND}} \cdot \underbrace{\sum_{m=0}^{|S|-|L|} C_{|S|-|L|}^m (-1)^m}_{=0} = 0$. Please see the complete derivation of the following formula.

$$
\begin{aligned}
\sum_{\emptyset \ne T \subseteq S} I_T^{\text{AND}} &= \sum_{\emptyset \ne T \subseteq S} \sum_{L \subseteq T} (-1)^{|T|-|L|} u_L^{\text{AND}} \\
&= \sum_{L \subseteq S} \sum_{T:L \subseteq T \subseteq S} (-1)^{|T|-|L|} u_L^{\text{AND}} - v_\emptyset^{\text{AND}} \\
&= \underbrace{u_S^{\text{AND}}}_{L=S} + \sum_{L \subseteq S, L \ne S} u_L^{\text{AND}} \cdot \underbrace{\sum_{m=0}^{|S|-|L|} C_{|S|-|L|}^m (-1)^m}_{=0} - v_\emptyset^{\text{AND}} \\
&= u_S^{\text{AND}} - v_\emptyset^{\text{AND}} = u_S^{\text{AND}} - v(\boldsymbol{x}_\emptyset)
\end{aligned}
\tag{24}
$$

Thus, we have $\forall \emptyset \ne S \subseteq N, u_S^{\text{AND}} = \sum_{\emptyset \ne T \subseteq S} I_T^{\text{AND}} + v(\boldsymbol{x}_\emptyset)$.

**(2) Universal matching theorem of OR interactions.**

According to the definition of OR interactions, we will derive that $\forall S \subseteq N, u_S^{\text{OR}} = \sum_{T:T \cap S \ne \emptyset} I_T^{\text{OR}}$, where we define $v_\emptyset^{\text{OR}} = 0$ (recall that in Step (1), we attribute the output on empty input to AND interactions).

Specifically, the OR interaction is defined as $I_T^{\text{OR}} = -\sum_{L \subseteq T}(-1)^{|T|-|L|}v_{N \setminus L}^{\text{OR}}$. Similar to the above derivation of the universal matching theorem of AND interactions, to compute the sum of OR interactions $\sum_{T:T \cap S \ne \emptyset} I_T^{\text{OR}} = \sum_{T:T \cap S \ne \emptyset}\left[-\sum_{L \subseteq T}(-1)^{|T|-|L|}v_{N \setminus L}^{\text{OR}}\right]$, we first exchange the order of summation of the set $L \subseteq T \subseteq N$ and the set $T : T \cap S \ne \emptyset$. That is, we compute all linear combinations of all sets $T$ containing $L$ with respect to the model outputs $v_{N \setminus L}^{\text{OR}}$ given a set of input variables $L$, i.e., $\sum_{T:T \cap S \ne \emptyset, T \supseteq L}(-1)^{|T|-|L|}v_{N \setminus L}^{\text{OR}}$. Then, we compute all summations over the set $L \subseteq N$.

In this way, we can compute them separately for different cases of $L \subseteq T \subseteq N, T \cap S \ne \emptyset$. In the following, we consider the cases (1) $L = N \setminus S$, (2) $L = N$, (3) $L \cap S \ne \emptyset, L \ne N$, and (4) $L \cap S = \emptyset, L \ne N \setminus S$, respectively.

(1) When $L = N \setminus S$, the linear combination of all subsets $T$ containing $L$ with respect to the model output $v_{N \setminus L}^{\text{OR}}$ is $\sum_{T:T \cap S \ne \emptyset, T \supseteq L}(-1)^{|T|-|L|}v_{N \setminus L}^{\text{OR}} = \sum_{T:T \cap S \ne \emptyset, T \supseteq L}(-1)^{|T|-|L|}u_S^{\text{OR}}$. For all sets $T : T \supseteq L, T \cap S \ne \emptyset$ (then $T \ne N \setminus S, T \ne L$), let us consider the linear combinations of all sets $T$ with number $|T|$ for the model output $u_S^{\text{OR}}$, respectively. Let $|T'| := |T| - |L|$, $(1 \le |T'| \le |S|)$, then there are a total of $C_{|S|}^{|T'|}$ combinations of all sets $T'$ of order $|T'|$. Thus, given $L$, accumulating the model outputs $u_S^{\text{OR}}$ corresponding to all $T \supseteq L$, then $\sum_{T:T \cap S \ne \emptyset, T \supseteq L}(-1)^{|T|-|L|}v_{N \setminus L}^{\text{OR}} = u_S^{\text{OR}} \cdot \underbrace{\sum_{|T'|=1}^{|S|} C_{|S|}^{|T'|}(-1)^{|T'|}}_{=-1} = -u_S^{\text{OR}}$.

(2) When $L = N$ (then $T = N$), the linear combination of all subsets $T$ containing $L$ with respect to the model output $v_{N \setminus L}^{\text{OR}}$ is $\sum_{T:T \cap S \ne \emptyset, T \supseteq L}(-1)^{|T|-|L|}v_{N \setminus L}^{\text{OR}} = (-1)^{|N|-|N|}v_\emptyset^{\text{OR}} = v_\emptyset^{\text{OR}}$.

(3) When $L \cap S \ne \emptyset, L \ne N$, the linear combination of all subsets $T$ containing $L$ with respect to the model output $v_{N \setminus L}^{\text{OR}}$ is $\sum_{T:T \cap S \ne \emptyset, T \supseteq L}(-1)^{|T|-|L|}v_{N \setminus L}^{\text{OR}}$. For all sets $T : T \supseteq L, T \cap S \ne \emptyset$, let us consider the linear combinations of all sets $T$ with number $|T|$ for the model output $u_S^{\text{OR}}$, respectively. Let us split $|T| - |L|$ into $|T'|$ and $|T''|$, i.e., $|T| - |L| = |T'| + |T''|$, where $T' = \{i | i \in T, i \notin L, i \in N \setminus S\}$, $T'' = \{i | i \in T, i \notin L, i \in S\}$ (then $0 \le |T''| \le |S| - |S \cap L|$) and $|T'| + |T''| + |L| = |T|$. In this way, there are a total of $C_{|S|-|S \cap L|}^{|T''|}$ combinations of all sets $T''$ of

order $|T''|$. Thus, given $L$, accumulating the model outputs $v_{N\setminus L}^{\text{OR}}$ corresponding to all $T \supseteq L$, then

$$\sum_{T:T\cap S\neq\emptyset,T\supseteq L}(-1)^{|T|-|L|}v_{N\setminus L}^{\text{OR}} = v_{N\setminus L}^{\text{OR}}\cdot\sum_{T'\subseteq N\setminus S\setminus L}\underbrace{\sum_{|T''|=0}^{|S|-|S\cap L|}C_{|S|-|S\cap L|}^{|T''|}(-1)^{|T'|+|T''|}}_{=0}=0.$$

(4) When $L \cap S = \emptyset, L \neq N \setminus S$, the linear combination of all subsets $T$ containing $L$ with respect to the model output $v_{N\setminus L}^{\text{OR}}$ is $\sum_{T:T\cap S\neq\emptyset,T\supseteq L}(-1)^{|T|-|L|}v_{N\setminus L}^{\text{OR}}$. Similarly, let us split $|T| - |L|$ into $|T'|$ and $|T''|$, i.e., $|T| - |L| = |T'| + |T''|$, where $T' = \{i|i \in T, i \notin L, i \in N \setminus S\}$, $T'' = \{i|i \in T, i \in S\}$ (then $0 \leq |T''| \leq |S|$) and $|T'| + |T''| + |L| = |T|$. In this way, there are a total of $C_{|S|}^{|T''|}$ combinations of all sets $T''$ of order $|T''|$. Thus, given $L$, accumulating the model outputs $v_{N\setminus L}^{\text{OR}}$ corresponding to all $T \supseteq L$, then $\sum_{T:T\cap S\neq\emptyset,T\supseteq L}(-1)^{|T|-|L|}v_{N\setminus L}^{\text{OR}} =$

$$v_{N\setminus L}^{\text{OR}}\cdot\sum_{T'\subseteq N\setminus S\setminus L}\underbrace{\sum_{|T''|=0}^{|S|}C_{|S|}^{|T''|}(-1)^{|T'|+|T''|}}_{=0}=0.$$

Please see the complete derivation of the following formula.

$$\begin{aligned}
\sum_{T:T\cap S\neq\emptyset}I_T^{\text{OR}} &= \sum_{T:T\cap S\neq\emptyset}\left[-\sum_{L\subseteq T}(-1)^{|T|-|L|}v_{N\setminus L}^{\text{OR}}\right]\\
&= -\sum_{L\subseteq N}\sum_{T:T\cap S\neq\emptyset,T\supseteq L}(-1)^{|T|-|L|}v_{N\setminus L}^{\text{OR}}\\
&= -\left[\sum_{|T'|=1}^{|S|}C_{|S|}^{|T'|}(-1)^{|T'|}\right]\cdot\underbrace{u_S^{\text{OR}}}_{L=N\setminus S} - \underbrace{v_\emptyset^{\text{OR}}}_{L=N}\\
&\quad -\sum_{L\cap S\neq\emptyset,L\neq N}\left[\sum_{T'\subseteq N\setminus S\setminus L}\left(\sum_{|T''|=0}^{|S|-|S\cap L|}C_{|S|-|S\cap L|}^{|T''|}(-1)^{|T'|+|T''|}\right)\right]\cdot v_{N\setminus L}^{\text{OR}}\\
&\quad -\sum_{L\cap S=\emptyset,L\neq N\setminus S}\left[\sum_{T'\subseteq N\setminus S\setminus L}\left(\sum_{|T''|=0}^{|S|}C_{|S|}^{|T''|}(-1)^{|T'|+|T''|}\right)\right]\cdot v_{N\setminus L}^{\text{OR}}\\
&= -(-1)\cdot u_S^{\text{OR}} - v_\emptyset^{\text{OR}} - \sum_{L\cap S\neq\emptyset,L\neq N}\left[\sum_{T'\subseteq N\setminus S\setminus L}0\right]\cdot v_{N\setminus L}^{\text{OR}}\\
&\quad -\sum_{L\cap S=\emptyset,L\neq N\setminus S}\left[\sum_{T'\subseteq N\setminus S\setminus L}0\right]\cdot v_{N\setminus L}^{\text{OR}}\\
&= u_S^{\text{OR}} - v_\emptyset^{\text{OR}}\\
&= u_S^{\text{OR}}
\end{aligned}$$

(25)

**(3) Universal matching theorem of AND-OR interactions.** With the universal matching theorem of AND interactions and the universal matching theorem of OR interactions, we can easily get $v(\boldsymbol{x}_S) = u_S^{\text{AND}} + u_S^{\text{OR}} = v(\boldsymbol{x}_\emptyset) + \sum_{\emptyset\neq T\subseteq S}I_T^{\text{AND}} + \sum_{T:T\cap S\neq\emptyset}I_T^{\text{OR}}$, thus, we obtain the universal matching theorem of AND-OR interactions.

$\square$

# E PRACTICAL VALUE

**Improving the performance of certain models in practice.** In fact, we find that our proposed loss function $\hat{\mathcal{L}}$, designed to eliminate non-convergent interactions, not only mitigates overfitting in DNNs (see Section 3.2) but also improves performance (*i.e.,* testing accuracy) in certain models. Specifically, Figure 6 shows both the testing accuracy achieved during normal training with cross-

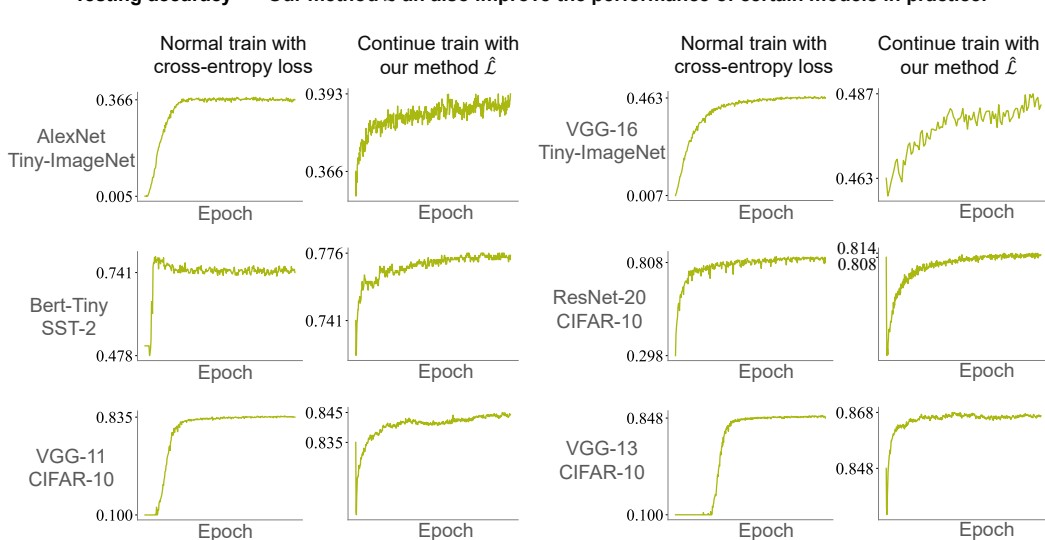

Figure 6: The proposed loss function $\hat{\mathcal{L}}$ successfully improved the performance of certain models in practice.

entropy loss and the testing accuracy after further training with our loss function $\hat{\mathcal{L}}$. *The results show that, for some models, our loss function yields a modest yet consistent performance gain.*

**Providing significant insights into identifying the precise point of early stopping in DNNs.** In the field of symbolic generalization, this paper is the first to establish a rigorous connection between a DNN's generalization power and the generalizability of interactions. Consequently, our explanation theory may provide a principled way to determine the optimal early stopping point in a DNN's training process by objectively explaining how the generalizability of encoded interactions evolves. Specifically, preliminary experiments in Figure 2 shows that the dynamics of all interactions (*i.e.,* $\mathbf{I}_{\text{pos}}^{(k)}$ and $\mathbf{I}_{\text{neg}}^{(k)}$) was closely aligned with the dynamics of convergent interactions (*i.e.,* $\mathbf{J}_{\text{pos}}^{(k)}$ and $\mathbf{J}_{\text{neg}}^{(k)}$) during both the normal training phase (when the training-testing loss gap is relatively small) and the overfitting phase (when the training-testing loss gap begins to widen). During the normal training phase, the generalizability of interactions gradually increases; however, once the DNN enters the overfitting phase, it begins to encode non-generalizable interactions (*i.e.,* high-order and mutually offsetting ones), which serve as a clear signal of overfitting. Therefore, *by tracking the evolution of interaction generalizability throughout the entire training process, one can stop training at the point where interaction generalizability peaks, offering guidance for early stopping from the perspective of the detailed inference patterns encoded by the DNN.*

## F  A SIGNIFICANT CONTRIBUTION TO SYMBOLIC GENERALIZATION

In the field of symbolic generalization, our method $\hat{\mathcal{L}}$ of proactively eliminating non-convergent interactions is, to our knowledge, the first to successfully increase the proportion of interactions that generalize to testing samples, which has long been regarded (Chen et al., 2024) as one of the most difficult challenges in this area. Specifically, we approximate the generalizability to testing samples by computing the ratio of interactions in a target DNN that converge to those in a baseline DNN trained directly on the testing set. Such convergent interactions can then be explained as primitive inference patterns inherently present in the test distribution. To evaluate this, we conducted experiments across multiple architectures: VGG-11, VGG-13, ResNet-20, and ResNet-32 on CIFAR-10; AlexNet and VGG-16 on Tiny-ImageNet; and BERT-Tiny, BERT-Medium, and BERT-Base on SST-2. For each architecture, we trained two DNNs with different random initializations, one serving as the target DNN $v$ and the other as the baseline DNN $v'$. With the baseline DNN trained on the testing

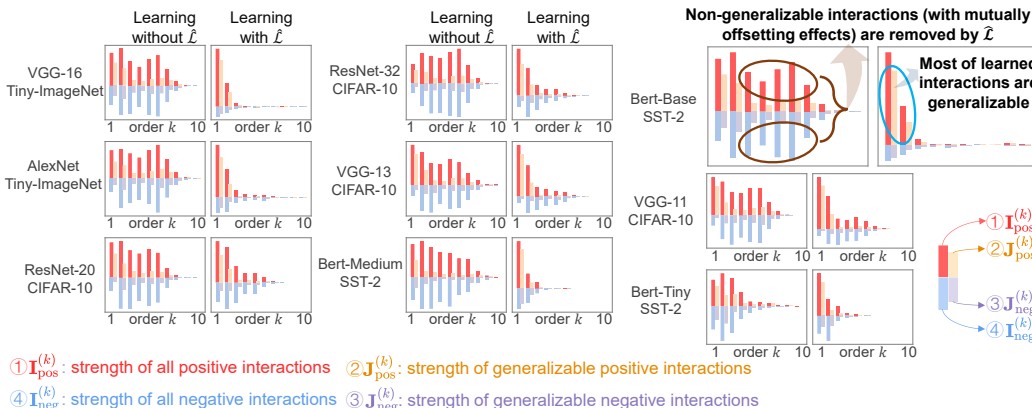

Figure 7: The proposed loss function $\hat{\mathcal{L}}$ successfully increases the proportion of interactions that successfully generalize to testing samples, **because we trained the baseline DNNs on testing samples in this experiment.** The distribution of interactions is computed by averaging the distribution across different samples.

set, the convergent interactions identified between the two networks can be regarded as interactions that generalize to testing samples.

Figure 7 shows that our method of eliminating non-convergent interactions increases the proportion of interactions that efficiently generalize to testing samples.

## G  EXAMPLES OF AND-OR LOGICAL MODELS EXPLAINING LLMS

In this section, to demonstrate that AND-OR logical models can faithfully explain large language models, we present examples showing that, given specific input prompts, AND-OR logical models can be constructed to faithfully explain both the DeepSeek-r1-distill-llama-8b (Guo et al., 2025) and Qwen2.5-7b (Bai et al., 2023) models, hereafter referred to as the DeepSeek and Qwen models, respectively. Figure 8 illustrates the AND-OR logical models used to explain DeepSeek and Qwen under different input prompts.

## H  EXPERIMENTS: ONLY OUR PROPOSED METHOD EFFECTIVELY REDUCES THE TESTING-TRAINING CROSS-ENTROPY LOSS GAP.

In this section, we conduct experiments to compare the two competing loss functions (*i.e.,* $\mathcal{L}_z$ and $\mathcal{L}_f$) introduced in Section 3.4 with our proposed loss function $\hat{\mathcal{L}}$, focusing on their effects on the cross-entropy loss gap between the training and testing sets. We trained VGG-11 and VGG-13 on the CIFAR-10 dataset.

As shown in Figure 9, only our method successfully reduced the training-testing cross-entropy loss gap, whereas the competing methods failed to achieve such an effect. This result demonstrates that our approach is uniquely effective in mitigating overfitting in DNNs.

## I  COMMON CONDITIONS FOR SPARSE INTERACTIONS

Ren et al. (2024a) have proved three sufficient conditions for the sparsity of AND interactions.

**Condition 1.** *The DNN does not encode extremely high-order interactions:* $\forall\, T \in \{T \subseteq N | |T| \geq M + 1\}$, $I_T^{\text{and}} = 0$.

Condition 1 is common because extremely high-order interactions usually represent very complex and over-fitted patterns, which are unlikely to be learned by a well-trained DNN in real scenarios.

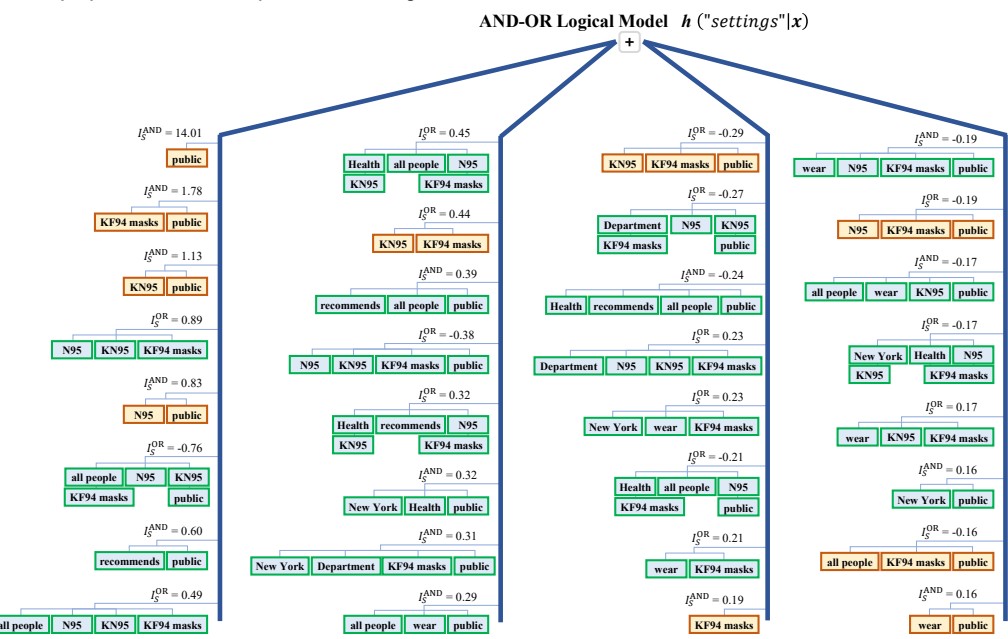

Figure 8: An example of AND-OR logical models constructed to faithfully explain the output scores of the DeepSeek model (top) and the Qwen model (bottom) on a single sample. A further example is presented on the next page.

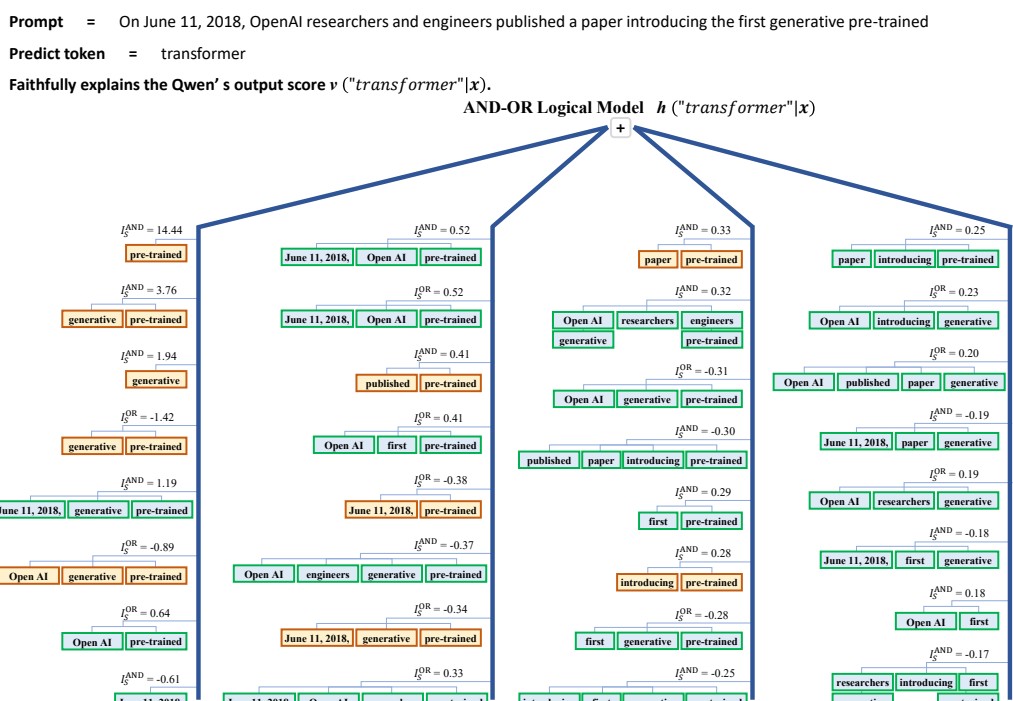

Figure 8: Another example of AND-OR logical models explaining the DeepSeek model (top) and the Qwen model (bottom) on a different sample.

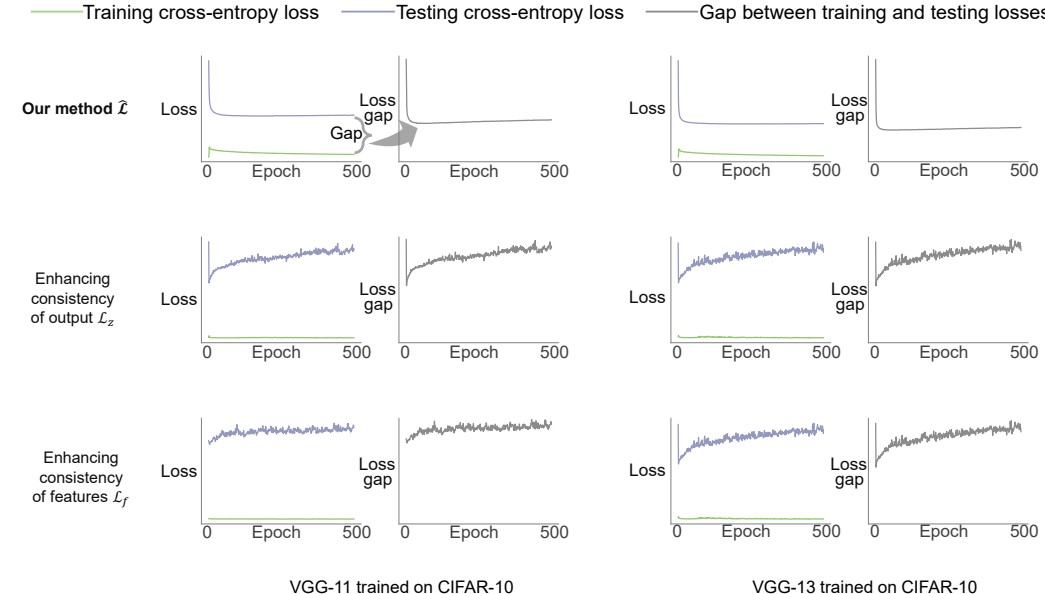

Figure 9: Only our method $\hat{\mathcal{L}}$ successfully reduced the training-testing cross-entropy loss gap, whereas the competing methods (*i.e.,* $\mathcal{L}_z$ and $\mathcal{L}_f$) failed to achieve such an effect.

**Condition 2.** *Let* $\bar{u}^{(k)} \stackrel{\text{def}}{=} \mathbb{E}_{|S|=k}[v(\boldsymbol{x}_S) - v(\boldsymbol{x}_\emptyset)]$ *denote the average classification confidence of the DNN over all masked samples* $\boldsymbol{x}_S$ *with* $k$ *unmasked input variables. This average classification confidence monotonically increases when* $k$ *increases:* $\forall\, k' \leq k,\ \bar{u}^{(k')} \leq \bar{u}^{(k)}$.

Condition 2 implies that a well-trained DNN is likely to have higher average classification confidence for less masked input samples.

**Condition 3.** *Given the average classification confidence* $\bar{u}^{(k)}$ *of samples with* $k$ *unmasked input variables, there is a polynomial lower bound for the average classification confidence with* $k'(k' \leq k)$ *unmasked input variables:* $\forall\, k' \leq k,\ \bar{u}^{(k')} \geq (\frac{k'}{k})^p\, \bar{u}^{(k)}$, *where* $p > 0$ *is a constant.*

Condition 3 suggests that the classification confidence of the DNN remains relatively stable even when presented with masked input samples. In real-world applications, the classification or detection of masked or occluded samples frequently occurs. As a result, a well-trained DNN typically develops the ability to classify such masked inputs by leveraging local information, which can be derived from the visible portions of the input. Consequently, the model should not produce a substantially reduced confidence score for masked samples.

## J  EXPERIMENTAL DETAIL

### J.1  TRAINING SETTINGS

• **Detailed settings of normal training.** In this paper, we trained various DNNs for different tasks. Specifically, for the image classification task, we trained VGG-11/13 on the CIFAR-10 dataset with a learning rate of $0.01$. We trained ResNet-20/32 on the CIFAR-10 dataset with a learning rate of $0.01$. We trained AlexNet and VGG-16 on the Tiny-ImageNet dataset with a learning rate of $0.01$. For the sentiment classification task, we trained the Bert-Tiny model, Bert-Medium model, and Bert-Base model on the SST-2 dataset with a learning rate of $0.01$. All DNNs were trained using the SGD optimizer (Robbins & Monro, 1951) with a momentum of 0.9.

For models trained on the CIFAR-10 dataset, we set the batch size to 256. Due to memory constraints, for models trained on the Tiny-ImageNet dataset, we used a batch size of 64. For models trained on the SST-2 dataset, we set the batch size to 64 (except for Bert-Tiny, which also used a batch size of 256).

- **Detailed settings for verifying the elimination of non-convergent.** To verify the effectiveness of our proposed loss function $\hat{\mathcal{L}}$ in eliminating non-convergent interactions, we followed the aforementioned normal training settings. Specifically, we first trained DNNs with the standard cross-entropy loss for 300 epochs. Subsequently, we further trained these DNNs with our proposed loss function $\hat{\mathcal{L}}$ for an additional 500 epochs and examined the resulting changes in the distribution of encoded interactions. Except for VGG-16, BERT-Base, and BERT-Tiny, which were trained for 100, 50, and 100 epochs, respectively, due to computational constraints, all other models were trained following the settings described above. For each network architecture, we trained two DNNs with differently initialized parameters, which served as the target DNN $v$ and the baseline DNN $v'$, respectively. We set the scalar weight $\lambda$ to 0.3 and kept the batch size unchanged from the standard training.

- **Detailed settings of verifying eliminating non-convergent interactions effectively reduce the training-testing cross-entropy loss gap.** Also, we first trained the DNNs with the standard cross-entropy loss for 300 epochs. We then continued training them with our proposed loss function $\hat{\mathcal{L}}$, using different values of $\lambda$, for an additional 500 epochs, and monitored the changes in the testing-training cross-entropy loss. For each network architecture, we trained two DNNs with differently initialized parameters, which served as the target DNN $v$ and the baseline DNN $v'$, respectively.

- **Detailed settings of comparing DNNs trained using our proposed loss function $\hat{\mathcal{L}}$ with DNNs trained on the two competing loss functions ($\mathcal{L}_z$ and $\mathcal{L}_f$).** we first trained the DNNs with the standard cross-entropy loss for 300 epochs. For training with loss terms that promote consistency of output ($\mathcal{L}_z$) and intermediate features ($\mathcal{L}_f$), we continued training the models for 500 additional epochs using a learning rate of 0.001. Again, we set $\lambda = 0.3$ and kept the batch size unchanged. For VGG-11, we empirically encouraged the consistency of intermediate features at the output of the 4th convolutional layer. For VGG-13, we enforced consistency at the output of the 5th convolutional layer. For ResNet-20, we promoted the consistency of intermediate features at the output of the 3rd residual block. For ResNet-32, we did so at the output of the 5th residual block. For each network architecture, we trained two DNNs with differently initialized parameters, which served as the target DNN $v$ and the baseline DNN $v'$, respectively.

## J.2 DETAILED SETTINGS OF TRAINING THE BASELINE DNNs

We can consider two types of interaction generalizability. (1) Across-sample generalizability, which refers to whether interactions learned from training samples can also be expressed on testing samples. (2) Across-model generalizability, which refers to whether an interaction is consistently encoded by different DNNs; for clarity, we refer to such interactions as convergent interactions. Moreover, because across-sample generalizability is difficult to measure directly, in practice it can be operationalized as a special case of convergent interactions, *i.e.,* those that converge to baseline DNNs trained on the testing samples. Such convergent interactions can therefore also be regarded as generalizable interactions.

Therefore, when assessing whether an interaction is *convergent* (*i.e.,* can be regarded as generalizable across DNNs), it suffices to check whether the interaction is simultaneously encoded by different DNNs, namely both the target DNN and the baseline DNN. To this end, we only need to train two DNNs with different random initializations on the same dataset. However, to evaluate whether interactions can *generalize to testing samples*, we further trained a baseline DNN directly on the testing set and examined whether the interaction was encoded by the baseline DNN. When the baseline DNN is trained on the testing set, the interactions it encodes can be regarded as generalizable to testing samples (*i.e.,* the baseline DNN also uses these interactions to classify testing samples). To ensure a fair comparison (*i.e.,* the same number of training samples for each model), we randomly split the training set of each dataset into two equal parts. One part was used as the training set and the other as the testing set. We then trained separate models on each subset and measured the generalization power of interactions accordingly.

## J.3 DETAILS ABOUT HOW TO CALCULATE INTERACTIONS FOR DIFFERENT DNNs

- **For image data in different image datasets,** since the computational cost of interactions was intolerable, we applied a sampling-based approximation method to calculate AND-OR interactions. Specifically, we considered the feature map after the low-layer as intermediate-layer features of

DNNs. We uniformly split the central region of each intermediate-layer feature (*i.e.,* we did not consider the pixel on the edges of an image) into $5 \times 5$ patches and selected the 10 patches with the highest L1 norms (*i.e.,* the brightest ones) to calculate interactions, and considered these patches as input variables for each intermediate-layer feature. We used $\mathbf{0}$ as a baseline value to mask the variables in $N \backslash T$.

• **For natural language data in SST-2 dataset,** we considered the outputs of the low-layer corresponding to input words as input features. We considered the embeddings corresponding to input features as input variables for each input sentence, and we randomly sampled 10 words, which must have a specific meaning and not be stop words, to calculate interactions. We used the average embedding over different input variables to mask the tokens in $N \backslash T$.

---

**Algorithm 1** Compute AND and OR Interactions and Select Salient Ones

---

**Require:** Deep neural network $v$, input sample $\boldsymbol{x} = [x_1, x_2, \ldots, x_n]^T$, set of indices $N = \{1, 2, \ldots, n\}$, small noise threshold $\zeta$, significance threshold $\tau$, convergence threshold $\epsilon$.

**Ensure:** AND interactions $I_T^{\text{AND}}$, OR interactions $I_T^{\text{OR}}$, and significant interaction sets $\Omega_{\text{sparse}}^{\text{AND}}$ and $\Omega_{\text{sparse}}^{\text{OR}}$.

1: Initialize learnable parameters $\{\gamma_L\}$ and $\{\delta_L\}$ for all $L \subseteq N$.
2: Compute baseline output $v(\boldsymbol{x}_\emptyset)$, where $\boldsymbol{x}_\emptyset$ is the masked sample with all variables removed.
3: Initialize previous loss $\mathcal{L}_{\text{prev}} \leftarrow \infty$.
4: **repeat**
5:    **for** each subset $L \subseteq N$ **do**
6:       Compute masked sample $\boldsymbol{x}_L$ by removing variables not in $L$.
7:       Compute network output $v(\boldsymbol{x}_L)$.
8:       Compute noise term $\delta_L$ constrained in $[-\zeta, \zeta]$, where $\zeta = 0.01 \cdot |v(\boldsymbol{x}) - v(\boldsymbol{x}_\emptyset)|$.
9:       Decompose $v(\boldsymbol{x}_L)$ into AND and OR components:
10:      $u_L^{\text{AND}} \leftarrow 0.5 \cdot (v(\boldsymbol{x}_L) - \delta_L) + \gamma_L$
11:      $u_L^{\text{OR}} \leftarrow 0.5 \cdot (v(\boldsymbol{x}_L) - \delta_L) - \gamma_L$
12:    **end for**
13:    **for** each subset $T \subseteq N$ **do**
14:       Compute AND interaction $I_T^{\text{AND}}$:

$$I_T^{\text{AND}} \leftarrow \sum_{L \subseteq T} (-1)^{|T|-|L|} u_L^{\text{AND}}$$

15:       Compute OR interaction $I_T^{\text{OR}}$:

$$I_T^{\text{OR}} \leftarrow - \sum_{L \subseteq T} (-1)^{|T|-|L|} u_{N \backslash L}^{\text{OR}}$$

16:    **end for**
17:    Compute current loss $\mathcal{L} \leftarrow \sum_{T \subseteq N} \left( |I_T^{\text{AND}}| + |I_T^{\text{OR}}| \right)$.
18:    Optimize parameters $\{\gamma_L\}$ and $\{\delta_L\}$ to minimize $\mathcal{L}$.
19:    Check for convergence: $|\mathcal{L} - \mathcal{L}_{\text{prev}}| < \epsilon$.
20:    Update previous loss: $\mathcal{L}_{\text{prev}} \leftarrow \mathcal{L}$.
21: **until** convergence
22: Select significant AND interactions:

$$\Omega_{\text{sparse}}^{\text{AND}} \leftarrow \{T \subseteq N : |I_T^{\text{AND}}| > \tau\}$$

23: Select significant OR interactions:

$$\Omega_{\text{sparse}}^{\text{OR}} \leftarrow \{T \subseteq N : |I_T^{\text{OR}}| > \tau\}$$

24: **return** $I_T^{\text{AND}}$, $I_T^{\text{OR}}$, $\Omega_{\text{sparse}}^{\text{AND}}$, and $\Omega_{\text{sparse}}^{\text{OR}}$.

---

## K    DETAILS TO EXTRACT THE SPARSEST AND-OR INTERACTIONS

A method is proposed (Li & Zhang, 2023b; Chen et al., 2024) to simultaneously extract AND interactions $I_T^{\text{AND}}$ and OR interactions $I_T^{\text{OR}}$ from the network output. Given a masked sample $\boldsymbol{x}_L$, Li & Zhang (2023b) proposed to learn a decomposition $v(\boldsymbol{x}_L) = u_L^{\text{AND}} + u_L^{\text{OR}}$ towards the sparsest interactions. The component $u_L^{\text{AND}}$ was explained by AND interactions, and the component $u_L^{\text{OR}}$ was explained by OR interactions. Specifically, they decomposed $v(\boldsymbol{x}_L)$ into $u_L^{\text{AND}} = 0.5 \cdot v(\boldsymbol{x}_L) + \gamma_L$ and $u_L^{\text{OR}} = 0.5 \cdot v(\boldsymbol{x}_L) - \gamma_L$, where $\{\gamma_L : L \subseteq N\}$ is a set of learnable variables that determine the decomposition. In this way, the AND interactions and OR interactions can be computed according to Theorem 1, *i.e.*, $I_T^{\text{AND}} = \sum_{L \subseteq T} (-1)^{|T|-|L|} u_L^{\text{AND}}$, and $I_T^{\text{OR}} = -\sum_{L \subseteq T} (-1)^{|T|-|L|} v_{N \setminus L}^{\text{OR}}$.

The parameters $\{\gamma_L\}$ were learned by minimizing the following LASSO-like loss to obtain sparse interactions:

$$\min_{\{\gamma_L\}} \sum_{T \subseteq N} |I_T^{\text{AND}}| + |I_T^{\text{OR}}| \tag{26}$$

**Removing small noises.** A small noise $\delta$ in the network output may significantly affect the extracted interactions, especially for high-order interactions. Thus, (Li & Zhang, 2023b) proposed to learn to remove a small noise term $\delta_T$ from the computation of AND-OR interactions. Specifically, the decomposition was rewritten as $u_L^{\text{AND}} = 0.5(v(\boldsymbol{x}_L) - \delta_L) + \gamma_L$ and $u_L^{\text{OR}} = 0.5(v(\boldsymbol{x}_L) - \delta_L) + \gamma_L$. Thus, the parameters $\{\delta_L\}$ and $\{\gamma_L\}$ are simultaneously learned by minimizing the loss function in Eq. (26). The values of $\{\delta_L\}$ were constrained in $[-\zeta, \zeta]$ where $\zeta = 0.01 \cdot |v(\boldsymbol{x}) - v(\boldsymbol{x}_\emptyset)|$.

**Algorithm of extracting AND-OR interactions.** The techinical details of computing $I_T^{\text{AND}}$ and $I_T^{\text{OR}}$ is provided in the following pseudocode in Algorithm 1.

