# OpenReview forum: "Towards the Mysteries of Convergent Interaction Representations through DNNs"
_ICLR.cc/2026/Conference — Submitted to ICLR 2026_

### Official Review · Reviewer_6RTT · 2025-10-27

**Soundness:** 2
**Presentation:** 2
**Contribution:** 2
**Rating:** 2
**Confidence:** 3

**Summary:**

This paper seeks to understand the generalization capabilities of Deep Neural Networks (DNNs) in terms of symbolic interactions that can be used to characterize the DNNs' behaviors. The authors do this across a wide variety of modalities and tasks. They show that "representation convergence of interactions" (defined as explanatory symbolic interactions that are shared across trained DNNs with different training seeds), can be used to predict DNN generalization capabilities. Where more convergent interactions tends to correlate with better model generalization capabilities. They then show that this predictive link is causal by training DNNs with an explicit objective to maximize convergent interactions and minimize divergent ones. They show that DNNs trained with this objective have better generalization capabilities.

**Strengths:**

- the authors are pursuing a promising/interesting direction of understanding generalization capabilities of DNNs through symbolic explanations.
- it's also a promising/interesting direction to use representational convergence (aka platonic representation hypothesis, anna karenina hypothesis) to understand network generalization.
- the causal experimental results are a novel, interesting way to reduce DNN generalization error during training.
- they show the value of their technique across a wide range of modalities and tasks

**Weaknesses:**

- generally difficult to work through material (especially for someone who is unfamiliar with the field of symbolic generalization)
- need to define "representation convergence of interactions" better in the introduction. That phrase does not speak for itself.
- if I am understanding correctly, the proposed method for determining "interaction logic" does not scale well (2^n where n is either vocabulary or context length?). And the method is restricted to fixed length inputs? Otherwise, how do you handle duplicate tokens in the context?
- when classifying whether there is a similar effect of a convergent interaction, merely requiring the sign to be the same is weaker evidence than requiring the difference in the values to exist within some tolerance. If one effect is 100 and the other is 0.1, this seems intuitively more "different" than values of -0.01 and 0.01, but their measure of "similarity" is based on sign alone in equation 5.
- need compare results in figure 4 to more classic regularization techniques (e.g. l2, dropout). It's unclear if results in figure 4 are merely an artifact of a training regime that encourages overfitting. They do look at l2 regularization in figure 5, but they only look at convergence of interactions rather than task performance. This renders figure 5 meaningless with respect to generalization capabilities.
- given that representations in DNNs are permutation invariant, the loss in equation 12 should be expected to have a deleterious effect on performance without increasing representational convergence.

**Questions:**

- on lines 202 and 203-204: does "convergent" imply "generalizable"? In what sense do you mean "generalizable"? It's possible to have consistent DNN representations that have poor generalization capabilities if generalization is defined with respect to out-of-distribution performance.
 - how sensitive are the symbolic interaction models to training data?
 - focusing on Figure 4, if you spent the compute used to ablate convergent interactions on model scale with aggressive regularization, how would the test loss compare to that generated from the ablation?
 - to what degree can your results be explained by the notion that the space of all DNN responses is far greater than the space of correct DNN responses?

---

### Official Review · Reviewer_n3NR · 2025-10-29

**Soundness:** 1
**Presentation:** 3
**Contribution:** 1
**Rating:** 2
**Confidence:** 2

**Summary:**

This paper tries to experimentally investigate the generalization power of DNNs. It approaches this problem by extracting rules (AND and OR statements with associated weights) from a DNN trained on the training-data  and compares the coefficients to extracted rules from a different DNN, called target DNN, trained on the test-data. The rules are constructed by first deciding on AND and OR statements and then computing the associated weights via a lasso regression for the respective DNN. The study first compares the similarity of the rules by comparing AND/OR statements and coefficients for a neural network during training to a target neural network trained on test. It finds that at first, during training, similarity increases and then, as the DNN starts to overfit, the similarity decreases again. In the second part, the paper tries to eliminate these non-convergent rules by adding a second loss over the similarity of those rules. It observes that the generalization gap decreases when the loss is activated and increases when it is deactivated. Furthermore, there is a small section on the effect of adding other similarity-measures between the training and target DNN on the generalization gap, where a similar behaviour is not observed

**Strengths:**

The main strength of the paper draws from the significance of the problem. Despite continuing empirical success of using neural networks, our understanding of generalization is still limited. From a better understanding one could hope to better design architectures, regularizations and diagnostics.The approach of comparing symbolic rules distilled from neural networks is also an interesting approach. The paper is also well written.

**Weaknesses:**

The main weakness of the paper is its conception of the generalization gap. A common way to define the generalization gap is the difference between the empirical risk and the true risk, where a test-set is used as an unbiased (assuming i.i.d) estimate of the true risk. This paper conflates the true risk and the empirical test risk. This is problematic as rules and quantities derived from the test-set might also not generalize to the true data-distribution: they might have overfit on the test-set. Another i.i.d test-set can provide an unbiased estimate, but this is apparently not done. Using these derived rules and quantities from the test-dataset to measure the relationship between the generalization gap during training (measured using the same test-dataset) is therefore problematic: the rules/quantities might be overfit on the test-set and whether this is a good indicator for general generalizability is not clear. Furthermore, the problem is more apparent in the interventional study. Adding a regularizing term based on the similarity to the rules derived from the test-set can leak information from the test-set to the training. That the generalization gap, as measured with the same test-set, can be reduced through this is understandable but the information drawn from this observation is limited.

Furthermore, the experimental validation of proposition 1 would benefit from a statistical analysis, as only plots and an imprecise description is provided. Also, comparing a single DNN to a single target DNN is also a limiting point. Having multiple target-DNNs, e.g. from multiple seeds, would enable a more detailed statistical analysis of this relationship, as the variance can not be investigated with a single run.
The paper also lacks important information in section 3.2., sets like $S_1$, $S_2$, $S_3$ remain vague. Also, it is not clear how exactly $v$ and $v^\prime$ are trained, as the text speaks of that both are trained using the loss $\hat{L}$, but $\hat{L}$ only mentions a single dataset $D$, but apparently $v$ is the DNN trained on the training dataset and $v’$ the DNN trained on the test-dataset. Also, equation $(8)$ does not specify what we are minimizing over, only the objective.

The section 3.4 and its appendix is also lacking details, a figure like figure 4 would help to understand the impact and whether the chosen similarity scores are correctly used. In the appendix it is specified that $\lambda=0.3$ is used to weight the two losses but it is not detailed how the authors arrived at this value.

A broader comparison to generalization concepts would also benefit the paper, e.g. the relationship between bias-variance tradeoff and the decomposition into convergent and non-convergent interactions.

**Questions:**

I think there is merit in the idea behind methodology proposed, however in its current form the paper is not ready for publication as I am questioning the way true and empirical risks over the test-set are conflated. I am open to hear more from the authors about the questions I raised.

How reproducible are the convergent interactions? So for a fixed target DNN, are the convergent interactions similar for multiple seeds for the baseline DNN?

The coefficients for the symbolic representations do not seem identifiable: it seems like you can encode the same function in different ways. How useful is it then to compute the positive/negative interactions if you decompose the interaction by k-th order. Can not the same function have different positive/negative interactions?

What is the motivation for differentiating positive and negative interactions?

How are the sets $S_1$,$S_2$,$S_3$ computed in section 3.2?

What is the result of setting $\lambda$ to $1$ for $\hat{L}$ (eq. 9)?

How is v’ trained using $\hat{L}$, as it should represent the DNN trained on the test-dataset. Are the roles reversed and $D$ is now the test-dataset?

---

### Official Review · Reviewer_pgVa · 2025-11-01

**Soundness:** 3
**Presentation:** 2
**Contribution:** 3
**Rating:** 6
**Confidence:** 2

**Summary:**

This paper explores the relationship between neural network generalization and the convergence of symbolic feature interactions across independently trained models. The authors build on recent “symbolic generalization” work, proposing that a DNN’s tendency to overfit is linked to the degree of representation convergence (i.e., how much overlap exists between the sets of feature interactions learned by different models trained on the same task). They find that 1) representation convergence and overfitting are strongly negatively correlated (meaning that overfitted models learn more idiosyncratic representations), and 2) increasing the convergence of interactions during training reduces overfitting. Empirical results on vision (CIFAR-10, Tiny-ImageNet) and language (SST-2) tasks validate these two propositions.

**Strengths:**

1. **Novel conceptual framing**. Provides a new perspective linking generalization to the convergence of symbolic feature interactions across models, which an interesting synthesis of interpretability and generalization theory.
2. **Clear empirical validation**. Both descriptive and interventional claims are tested systematically with real models and tasks.
4. **Methodologically careful**. Experiments span multiple architectures (VGG, ResNet, AlexNet, BERT) and both vision and language domains, suggesting breadth.
5. **Potentially useful insight**. The finding that interaction convergence peaks before overfitting could inspire new early stopping or regularization criteria.

**Weaknesses:**

1. **Modest performance gains**. The experimental gains are relatively modest, so the practical value of these insights seems a bit limited.
2. **Somewhat opaque exposition**. The exposition can be a bit hard to follow, and the core intuitions are buried under dense symbolic notation, making it a bit difficult for non-specialists to follow.
3. **Mechanistic plausibility is unclear**. The notion of “interactions” extracted via the AND-OR logical model seems a bit narrow and may not correspond to meaningful internal features in real networks, which limits interpretability credibility.

**Questions:**

1. Why should convergence across models necessarily imply generalizability, rather than merely inductive bias similarity?
2. How do these extracted “AND/OR interactions” relate to conventional interpretability artifacts (e.g., feature maps, attention heads, circuits)?
3. Have the authors tested whether the effect persists in larger, more modern architectures (e.g., ViT, GPT-like models)?

---

### Official Review · Reviewer_yQD2 · 2025-11-04

**Soundness:** 3
**Presentation:** 2
**Contribution:** 2
**Rating:** 2
**Confidence:** 3

**Summary:**

This paper conducts an experimental analysis on whether removing non-convergent interactions prevents overfitting in popular vision architectures. Starting from the studies of Ren et al. on Harsanii interactions, the idea is to find if any correlation exists between the difference of train and test loss and the nature of the interactions (convergent or non-convergent). The authors then show how to provide a loss that steers the student model towards less overfitting when promoting convergent interactions. This technique is compared to other popular regression methods for knowledge distillation.

**Strengths:**

The idea is sound and timely, comparing the faithful explanations (derived as interactions) with generalization is a promising venue. The merit of the authors is showing that, in some experimental contexts, keeping convergent interactions (that are those shared by two distinctly trained models) aid focusing on more salient features, giving more generalization powe.

I can see that authors' analysis has many intersections with shortcuts and spurious correlation.
In essence, if a model relies on misleading parts of the input -- in this case non-convergent interactions, it is likely that it suffers poorer generalization on test and may induce catastrophic errors in out-of-distribution. The idea of using Harsanii interactions is valid, since it focuses on explanations at input-level and can be used as a natural tool to highlight model shortcuts.

**Weaknesses:**

However, the fact that authors only compare with a small literature of XAI, mostly related to few works on Harsanii interactions, reduces the scope of the paper. I would have preferred the authors comparing also to the literature in shortcut learning, which is very relevant for what they have in mind. This also means that natural competitors should be taken from that line of research. I cannot see why knowledge distillation are proper competitors as, by purpose, they try to make student models more similar to teachers, which is not the goal of this paper.

There some things that are unclear to me by reading the paper:
1. Plots of convergent interactions are a bit small and require some scrutiny to distinguish convergent/non-convergent interactions. Would have it been better to report a ratio of convergent ints/all ints. This should be fixed for clarity sakes. Also, reporting the numbers somewhere would be ideal.
2. I doubt that the way of measuring convergent interactions could be done by only considering two models. In principle, the notion should account for an ensemble of predictors that equally fit the task. In fact, it is possible that two models, by chance, could learn almost the same interactions (giving ratio 1) or completely different interactions (ratio 0). For this, the choice of what is convergent and not would highly depend on the trained models, and so there is not a proper way to tell if one interaction is truly convergent.
3. Also from above, what prevents your loss function to learn exactly the same interactions of the teacher? I can imagine that one trivial solution to the training loss would be learning two identical models, but this goes against what convergent interactions are opposed to mean
4. Also, it seems a bit unnatural that forcing two models to be similar by interactions makes the model focusing on more salient features. I guess this would not avoid watermarks or shortcuts in the model, possibly giving less generalization.
5. This makes me wonder how large is the generalization error measured in Figure 4? Can you report also training/test errors to get an idea of what's happening?

**Questions:**

I asked questions in the box above. I believe this is an interesting work but needs to draw more connections to existing literature (which would increase the impact of the work) and clarify some aspects of the empirical analysis.

I am willingly to revise my score upon clarifications on the weaknesses.

---

### Meta-Review · Area_Chair_Rncc · 2026-01-05

**Summary:**

**Summary of Contribution** \
The paper investigates the link between neural network generalization and the convergence of learned feature interactions across independently trained models, showing that overfitting is associated with non-convergent, idiosyncratic interactions. It further demonstrates that explicitly encouraging interaction convergence through an additional training loss reduces the generalization gap.

**Summary of Concerns** \
All reviewers appreciated the novel analysis of experimentally studying generalisation from an interpretability perspective.
However, all reviewers raised important concerns with the **clarity** of the presentation, lacking proper definitions and running examples to explain the underlying concepts; with the **quality / soundness** of the experimental methodology, as requiring a broader analysis of the bias-variance tradeoff (Reviewer n3NR), missing important baselines for comparisons from the XAI literature and regularization techniques aimed at increasing generalization (Reviewer yQD2, 6RTT) and reporting the statistical significance of the results. Additionally, Reviewer pgVa raised important concerns about the scope of generality and significance of the proposed analysis (**significance**). This is indeed limited to AND-OR interaction rules which might not capture the underlying expressivity of deep neural network behaviour. Moreover, the practical improvements appear modest.

All concerns remain unaddressed, as authors didn’t engage during the rebuttal.

**Decision** \
Major clarity, quality and significance concerns remain unaddressed with the paper and therefore recommend for its rejection.

**Reviewer Concerns:**

All concerns were not addressed.

**Reviewer Scores:**

All reviewers would have kept their score.

---

### Decision · Program_Chairs · 2026-01-26

Reject